**Special Issue "Multi-risk assessment in the Andes region",**

**Journal "Natural Hazards and Earth System Sciences"**

# Insights into the Development of a Landslide Early Warning System Prototype in an Informal Settlement: the Case of Bello Oriente in Medellín, Colombia

Christian Werthmann[1], Marta Sapena[2], Marlene Kühnl[3], John Singer[4], Carolina Garcia[1], Tamara Breuninger[5], Moritz Gamperl[5], Bettina Menschik[5], Heike Schäfer[1], Sebastian Schröck[6], Lisa Seiler[1], Kurosch Thuro[5], Hannes Taubenböck[7]

[1]Leibniz Universität Hannover, Institut für Landschaftsarchitektur, Herrenhäuserstrasse 2a, 30419 Hannover, Germany
[2]German Aerospace Center (DLR), German Remote Sensing Data Center (DFD), Münchnerstrasse 20, 82234 Weßling, Germany [3]Company for Remote Sensing and Environmental Research (SLU), Kohlsteiner Str. 5, 81243 München, Germany
[4]AlpGeorisk, Lorenz-Hübner-Str. 15, 86609 Donauwörth, Germany
[5]Technische Universität München, Lehrstuhl für Ingenieurgeologie, Arcisstraße 21, 80333 München, Germany
[6]Technische Hochschule Deggendorf (THD), Technologie Campus Freyung (TCF), Institut für Angewandte Informatik, Grafenauerstr. 22, D-94078 Freyung, Germany
[7]German Aerospace Center (DLR), German Remote Sensing Data Center (DFD), Münchnerstrasse 20, 82234 Weßling, Germany; Institute for Geography and Geology, Julius-Maximilians-Universität Würzburg, 97074 Würzburg, Germany

*Correspondence to*: Christian Werthmann (werthmann@ila.uni-hannover.de)
ORCID ID: 0000-0002-8275-1558

**Abstract.** The global number of vulnerable citizens in areas of landslide risk is expected to increase due to the twin forces of climate change and growing urbanization. Self-constructed or informal settlements are frequently built in hazardous terrain such as landslide-prone slopes. They are characterized by high dynamics of growth, simple construction methods, strong social dynamics, and are exposed to unsteady political approaches. Landslide Early Warning Systems (LEWS) can contribute to decrease their vulnerability, but precise, affordable and culturally integrated LEWS need to be further developed. In this paper, we present a four-year living lab research project called Inform@risk that aimed to develop a LEWS prototype in the neighborhood of Bello Oriente, located in the urban-rural border of Medellín, Colombia. Its research team is composed of landscape architects, geo-engineers, remote sensing and geo-informatic experts. The research team collaborated with a multitude of stakeholders: civil society, private enterprises, non-governmental agencies and various branches of government. A preliminary LEWS with the last functionalities still to be developed has been designed, implemented and handed over to the government. It has entered a test and calibration phase (i.e., warning thresholds development, procedures for warning and alert dissemination through the sensor system), which is on hold due to legal constraints. First findings indicate that the integrative

development of technical aspects of a LEWS in informal settlements can be challenging, but manageable; whereas, the social and political support is beyond the control of the designer. Steady political will is needed to increase technical capacities and funding of the operation and maintenance of an increased amount of monitoring equipment. Social outreach has to be continuous in order to inform, train, maintain trust and increase the self-help capacities of the often rapidly changing population of an informal settlement. Legal requirements for a transfer of academic research projects to municipal authorities have to be clear from the start. Satisfying replacement housing options for the case of evacuation have to be in place in order to not lose the overall acceptance of the LEWS. As political will and municipal budgets can vary, a resilient LEWS for informal settlements has to achieve sufficient social and technical redundancy to maintain basic functionality even in a reduced governmental support scenario.

Keywords: landslides, early warning systems, risk, natural hazard, informal settlements, living lab

**Copyright Statement** will be included by Copernicus

## 1 Introduction

The global number of vulnerable people in areas of landslide risk is expected to increase due to the twin forces of climate change (IPCC, 2023) and rising urbanization rates (Taubenböck et al., 2012). For example, in the Andes region, about 1.4 million households are exposed to natural hazards (Peduzzi et al., 2009; UN-Habitat, 2015; UN-Habitat, 2016). Citizens living in informal settlements (neighborhoods that are constructed outside of governmental planning regulations) are disproportionally affected. Alone in the Colombian city of Medellín, 81% of the 290,000 residents exposed to landslide hazard dwell in substandard housing areas (Kühnl et al., 2021). And this is not only true for the Medellín example: Müller et al. (2020) showed that informal settlers tend to settle or have to settle in highly exposed landslide-prone locations. This exposure follows the global trend of socioeconomically vulnerable populations being disproportionally affected by the effects of climate change. Therefore, the sixth IPCC AR6 Synthesis Report (2023) stresses the urgency to prioritize issues of equity, climate justice, social justice, and inclusion when it comes to mitigation and climate resilient development.

The fundamental goal of risk reduction, as for example articulated in the Sendai Framework for Disaster Risk Reduction 2015-2030 (UNISDR, 2015), is for instance to be achieved by means of Early Warning Systems (EWSs), in our case against landslide hazards. This is a scientific and technical challenge on the one hand and a social challenge on the other. In this paper, the goals, the technical-social challenges and experiences with the implementation of a Landslide Early Warning System (LEWS) are presented; a possible way between abstract goals and local implementation in the field of risk reduction is to be shown. The presented findings follow the four-level structure of EWSs of UNISDR (UNDRR, 2023). The city of Medellín was chosen as a case study, because of a large number of residents (over 230,000) living in landslide-prone informal settlements (Kühnl et al., 2021), a pro-active government with a tracking history of innovative urban improvement (Drummond et al., 2012), and the existence of a unique early warning project in Latin America (SIATA: Sistema de Alerta Temprana del Valle de Aburrá). In

Medellín, a resettlement of citizens out of landslide hazard zones is infeasible given the large numbers of people at-risk and lack of urban land in low hazard areas in the city.

In cases, where resettlement or landslide mitigation efforts are infeasible or only partially effective LEWS can decrease human vulnerability, especially in informal settlements characterized by informal governance, insecure tenure, substandard housing and infrastructure conditions (Sultana and Tan, 2021). This is one of the reasons why EWSs are considered key tools for reducing disaster risks and supporting climate adaptation (UNFCCC, 2022). Consequently, several integrated LEWSs have

75 been implemented throughout the world in recent years (Fathani et al., 2016, Fathani et al., 2023; Peters et al., 2022). Based on these experiences the international standard ISO 22327 (2018), which provides a framework for the implementation of community-based LEWSs, has been developed. This general framework however needs to be tailored to the specific situation of informal settlements.

In the Aburrá valley, where Medellin is located, the regional environmental agency *Area Metropolitana Valle de Aburrá*

(AMVA) operates the applied research project SIATA. SIATA monitors environmental conditions like weather, hydrology, air quality, and seismic activity, among others. SIATA develops forecasts and can send out early alerts to AMVA and Medellin's risk management agency DAGRD, which is responsible for response on the ground. DAGRD, AMVA and SIATA collaborate to monitor active landslides that compromise relevant infrastructure in specific locations, but there is a general need for monitoring large high hazard areas with socially integrated EWSs. This applies especially to the informal

settlements that are located in the high hazard slopes of the rural-urban border of Medellín, where most population is highly vulnerable and therefore high risk prevails. Therefore, the research project Inform@Risk sought to develop an integrated LEWS in informal settlements with a socio-technical approach following a living lab methodology. As a learning and test case, an informal neighborhood exposed to landslides at the fringes of the Colombian city of Medellín was selected. Based on literature review and discussion with local experts, seven goals were set to develop an integrated LEWS: 1) precise 2)

affordable 3) socially integrated 4) multi-sectoral 5) socio-spatially integrated 6) multi-scalar and 7) replicable. In detail, these goals mean:

1) precise

Existing LEWSs in the Andes region rely on calculated probabilities based on the amount of rainfall in relation to previous

events (Sirangelo and Versace, 1996; Segoni et al., 2009). In several cities, including Medellín, geosensors that allow a higher precision of monitoring earth movements in real time are placed in active mass movements (Michoud et al., 2013), especially in areas with critical infrastructure (aqueducts, water tanks, oil pipes, main roads, etc.). In contrast, the project seeks to develop a self-learning automatic monitoring system also in non-active mass movement zones. The system should consist of sensors monitoring soil movement and water pressure, and use neuronal networks to increase the specificity of warnings and warning

100 times.

**2) affordable**

Precise landslide monitoring systems are available, but they can be cost-intensive. Therefore, the goal is to develop a more precise LEWS in accordance with the financial means of an emerging economy such as in Colombia. Actual affordability for individual cases will depend on the available budget of the stakeholder in charge. In our case, the preliminary test of affordability is, if the city of Medellín is able and willing to afford and replicate the prototype. General affordability shall be increased in materials, installation and maintenance by using cost-effective Internet of Things solutions for the sensor system (e.g. Prakasam et al., 2021; Esposito et al., 2022).

**3) socially integrated**

An integrated LEWS needs to be accepted, jointly developed with and supported by the residents of vulnerable areas (Basher et al. 2016; ISO 22327, 2018; Baudoin et al., 2016; Marchezini et al., 2018). Within the setting of the living lab, we seek to actively engage at-risk residents of the informal settlement in the design of the LEWS and its training activities.

**4) multi-sectoral**

An integrated LEWS relies on the successful collaboration of a variety of actors: residents, authorities, academic researchers, social organizations, entrepreneurs and non-governmental agencies (UNEP, 2012; García and Fearnley, 2012; Thapa and Adhikari, 2019; Alcántara-Ayala and Garnica, 2023). Thus, the design seeks the eye-to-eye collaboration with the identified actors in order to develop a fully integrated LEWS. It specifically tries to connect the civil society organizations and the residents of the informal neighborhood with the responsible risk management authorities.

**5) socio-spatially integrated**

The socio-spatial integration of a LEWS is especially relevant for informal neighborhoods characterized by strong social dynamics and alternative governance models (Brillembourg et al., 2005; Padilla et al., 2009). The constructed technical components of the LEWS should not be incomprehensible technical objects in the neighborhood. They should through a co-creative development process be familiar and comprehensible to the residents, serve as a natural reminder of landslide risk, increase the quality of public spaces in their daily use, the acceptance of the technical system and decrease vandalism.

**6) multi-scalar**

The research aims to gain and connect the knowledge of risk management on the scale of the neighborhood, the district level and the whole city. The consideration of these three scales should combine the specific situation of the particular neighborhood with the more general situation of the district and the whole city.

**7) replicable**

Direct transference of a successful approach from one informal settlement to another can be counterproductive as individual conditions in informal settlements do vary widely (Perlman, 2010). The gained knowledge in the development of the LEWS should yield process knowledge (not what-to, but how-to) that can be adapted to other locations with similar challenges.

This paper aims to show one of many possible paths from global risk reduction goals to local implementation. It shows the
developed socio-technical implementation path and reflects on the experiences made. The remainder of this paper is organized as follows: In section 2 the methodology is introduced, followed in section 3 by findings that are considered preliminary as the socio-technical LEWS is running in a test phase at the moment, and still has to prove its durability over longer time periods. In section 4 conclusions are drawn and section 5 ends this paper with a final statement.

## 2 Methodology and Process

The research project follows a living lab methodology, which has been applied in many different research contexts, but has internationally gained increasing relevance for research in the sustainable transformation of cities (Hossain et al., 2019; von Wirth et al., 2019). As the profound understanding of complex socio-technical processes plays a key role, the living lab methodology has been chosen for the development of an integrated LEWS in an informal settlement.

There is no uniform definition of living labs, because all have different goals and ways of working. However, a common characteristic is the horizontal partnership between scientists, various institutions and citizens, working together in a bounded space for intentional collaborative experience for a study case (Voytenco et al., 2015). Living Labs promise to produce a broader diversity of solutions as unexpected results are integral to the method (Schneidewind, 2014). Thereby, the living lab methodology is understood as a learning process, where all stakeholders are actively involved from the initial stage to jointly
identify challenges and different mind-sets. Transformation can be co-created and prototypes can be developed, which are then tested and improved in a real live environment. While the methodology can produce tailored innovation for social systems and governmental structures (Yasuoka et al., 2018), the transferability of its gained knowledge is context specific and varies. The more the living lab is dependent on its specific physical, social, and political context, the less transferable  its results are. In some cases, only patterns that might serve as useful guides for similar cases can be deduced (Schneidewind, 2014).

The living lab approach in this study integrates the following basic stakeholders: 1) scientific institutions, 2) residents, 3) governmental agencies, 4) local civil-society and non-governmental organizations (CBO's and NGO's), and 5) private companies (Fig. 1).

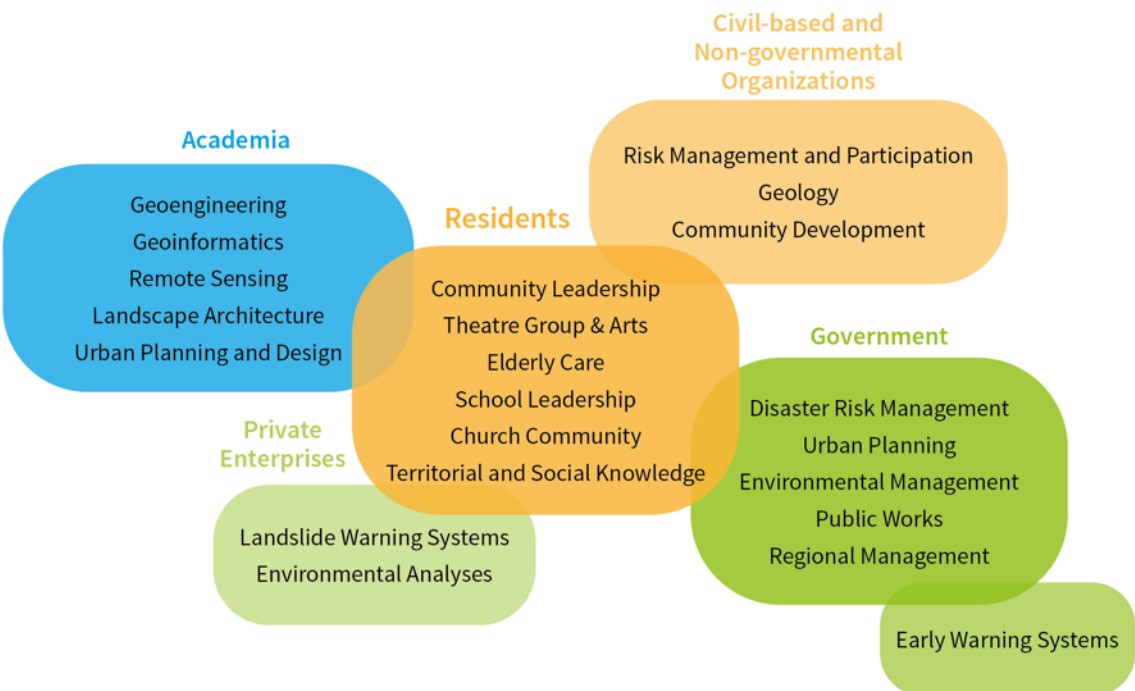

**Figure 1.** Capacities of actors in the research project. The composition is based on key actors in early warning systems (UN-ISDR, 2006; Fathani et al, 2016) and adjusted to the participating stakeholders in Medellín.

The living lab was structured into three phases: 1) investigation, 2) transformation and 3) implementation (Fig. 2). In order to gain an understanding of the goals of the actors, the instruments of action, the technical, social and political preconditions as well as the interfaces between the multi- and transdisciplinary actors, intensive communication is required. Throughout, monthly meetings of all stakeholders in Medellín, quarterly progress meetings of the scientific teams, as well as numerous workshops with residents and local experts were held. Extensive field work was pursued in form of physical, geological and geo-electric surveys, drillings, drone flights, community and expert workshops, meetings, interviews, questionnaires, and two representative surveys. External as well as internal events shaped the individual outcome of the study. Figure 2 gives a detailed overview of the research process in chronological order.


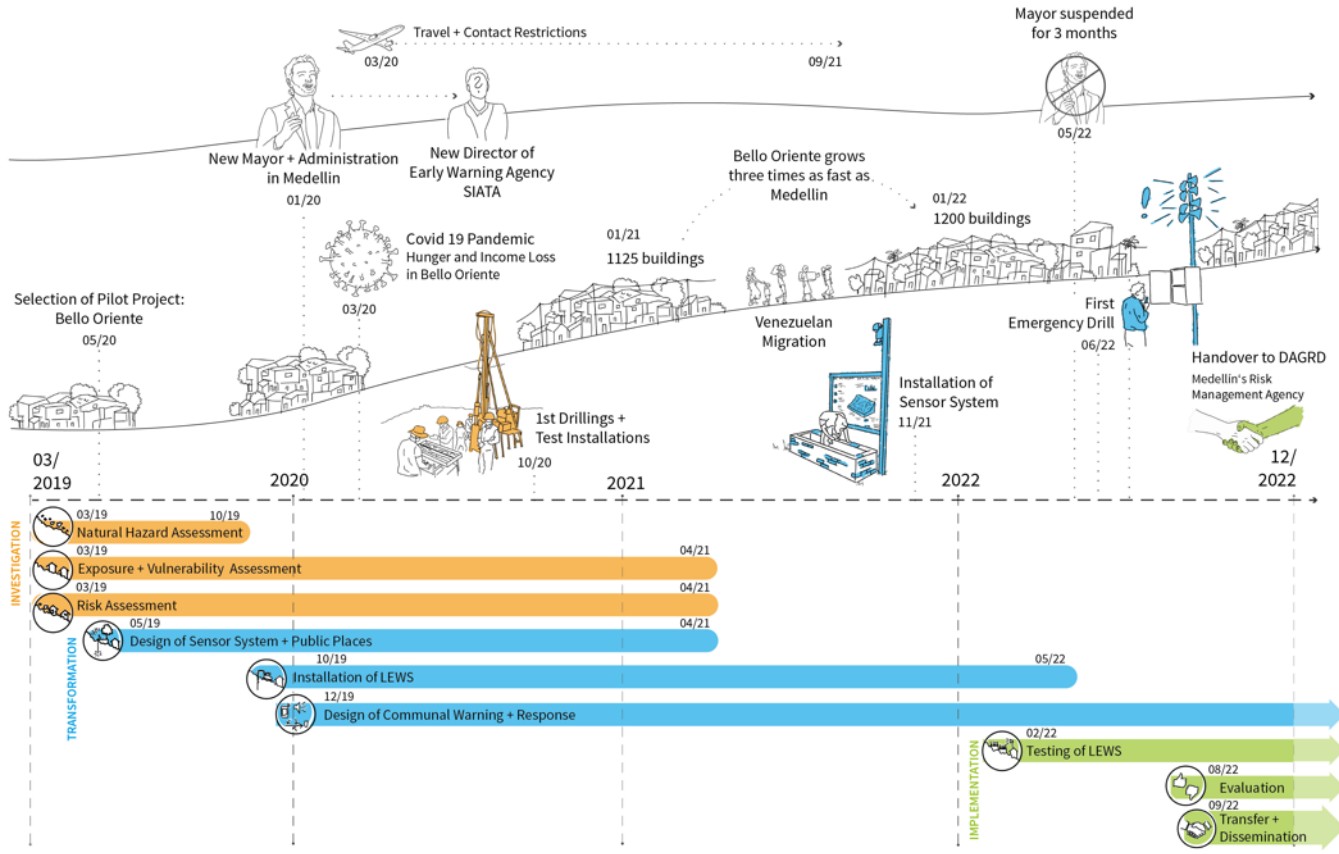

**Figure 2.** Time line of the living lab approach in relation to external and internal events that impacted the research.

In the first investigative phase, hazard and vulnerability assessments to landslides were conducted on the city, district, and neighborhood level of Medellín. For the selection of a pilot-neighborhood, 16 informal settlements with high landslide risk were examined based on 20 qualitative and quantitative technical, social and administrative parameters, such as risk, type of

past mass movements or willingness of the community to participate. Finally, the informal neighborhood of Bello Oriente on the northeastern fringe of Medellín (Fig. 3) was selected as case study based on several factors, being the four most critical:

1. landslides have recently occurred in the neighborhood with a consequent risk awareness of the population
2. the community voluntarily embraces the pilot project
3. the community is well organized and has experience in working with non-profit organizations
4. field work can be performed in an acceptably safe environment

Figure 3 illustrates the pilot-neighborhood. It shows the steep terrain in the upper reaches of Bello Oriente as well as the location of buildings and their usages as recorded based on drone data.


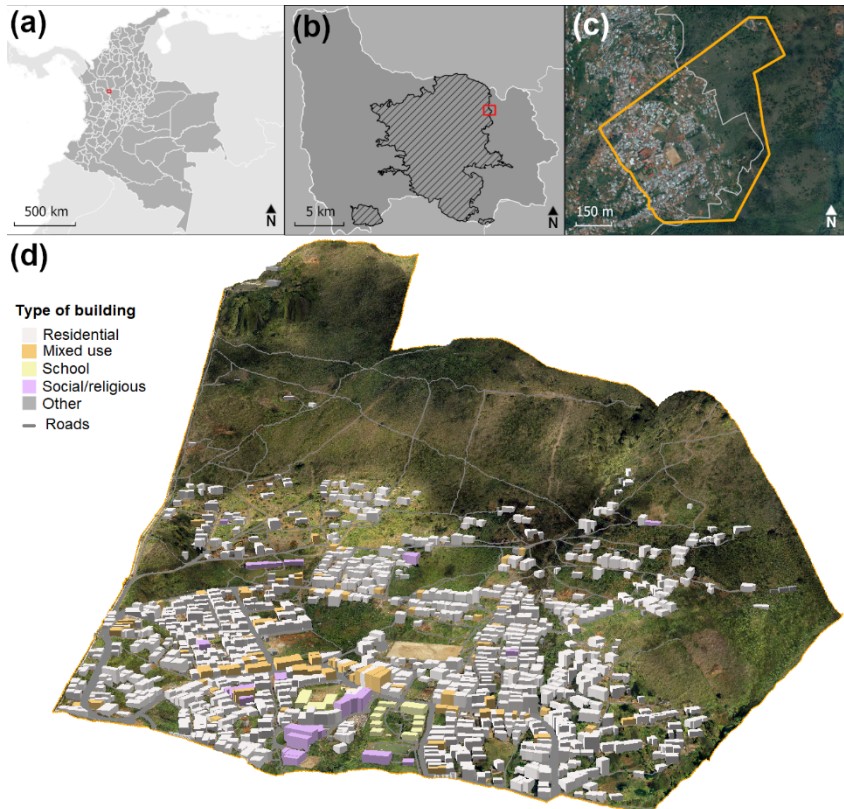

**Figure 3.** (a) Location map of Medellín within Colombia; (b) location map of the pilot-neighborhood Bello Oriente within the municipality of Medellín (white boundary) and city of Medellín (dashed area); (c) boundary of the project area within Bello Oriente, situated at the urban-rural border of the city,in the background the orthophoto of the city from 2019 (Alcaldía de

Medellín, 2023); and (d) 3D model of the upper part of Bello Oriente based on a 2021 drone flight.

The second transformative phase was characterized by the design of the LEWS and its subsequent installation in Bello Oriente. A two-pronged approach of landslide monitoring by a technical sensor system and residents was developed leading to multiple warning and dissemination options that will be explained in more detail later (Fig. 4). In general, both warning chains report

to the risk management authority of the city of Medellín. There, the information is processed to different warning types and communicated via various channels. From this phase onwards, the research project had local engineers from the project working constantly in the field.

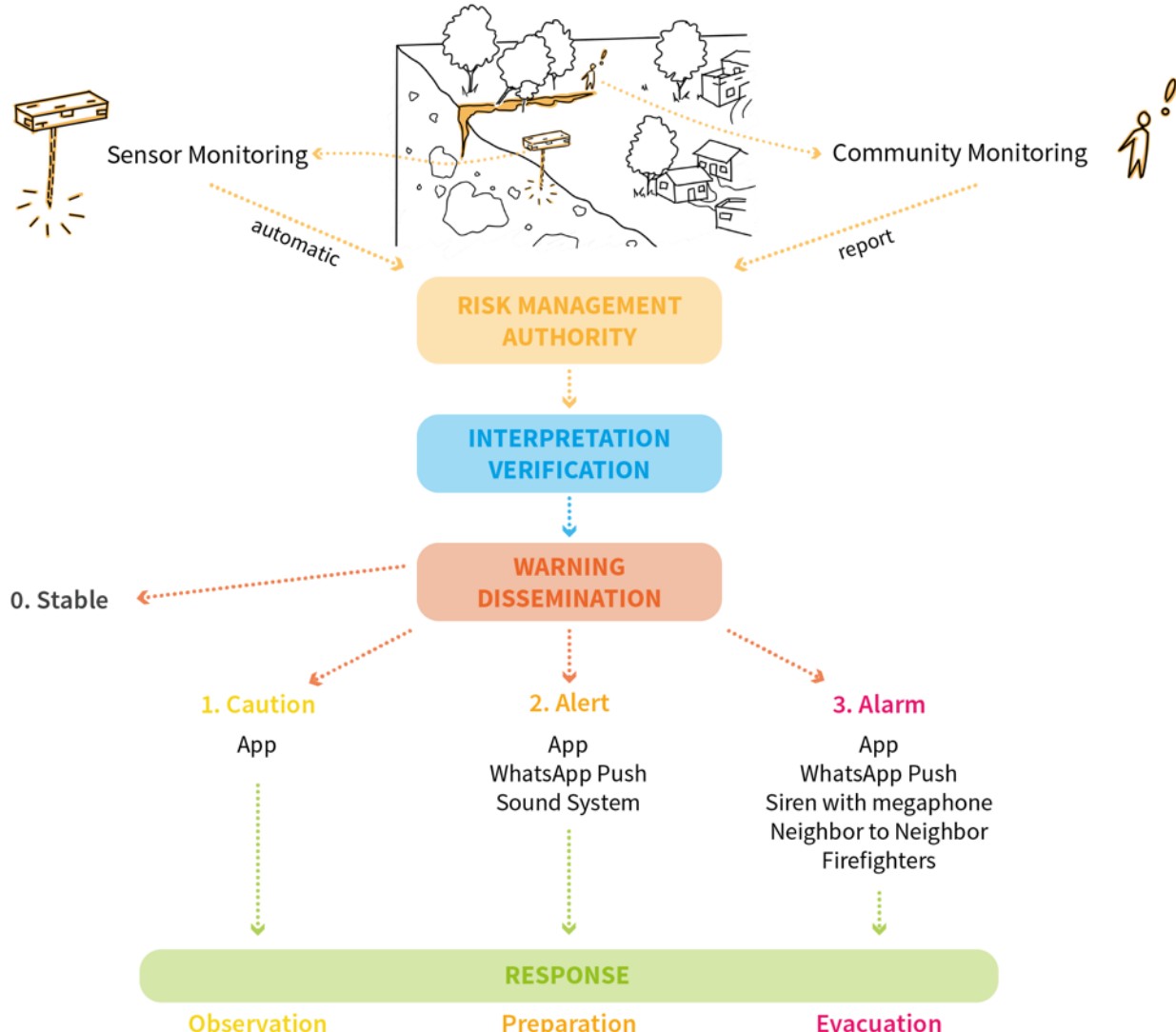

**Figure 4.** Landslide monitoring and warning design.

The implementation, the third phase, was characterized by community training and technical improvement. It concluded with the handover of a preliminary LEWS with the last functionalities still to be developed (i.e., warning thresholds development, procedures for warning and alert dissemination through the sensor system) to the disaster risk management authorities of Medellín at the end of 2022. After the handover, a reduced research team continued in 2023 with the calibration of the sensor components and the further development of a mobile application as part of a one-year development and training phase to test the inclusion of the prototype into the disaster risk management practices of the municipality and further evaluate its

usefulness. However, the continuation of this process has been put on hold as the legal constraints to donate the monitoring equipment to the municipality have not yet been resolved.

## 3 Preliminary Findings

### 3.1 Landslide Risk Analysis

### 3.1.1 Landslide Risk at City Scale

The city of Medellín is located in the Aburrá valley with steep slopes on both sides. By combining heterogeneous geodata sources - official geodata of the city and remote sensing - it becomes possible to map and analyze both, the topography and the exposed landslide zones as well as the settlement structures, their development, and the amount of people living in specific

parts of the city. This topographical situation in combination with heavy rain fall events and the increasing population numbers makes many of the 2.32 million inhabitants of Medellín exposed to landslide hazards due to their residential location. On the one hand, the landslide hazard map from the official land use plan (Plan de ordenamiento territorial, POT 2014), which includes the zoning of landslide hazards and combines all available risk-related maps, mass movement inventories, geotechnical and slope stability studies, and local knowledge (Alcaldía de Medellín, 2014), was used to map landslide susceptibility areas across

entire Medellín. These are areas of slopes mostly steeper than 17° predominantly formed by debris flow deposits and Medellín Dunite. On the other hand, census data from 2018 was used for a very-high-resolution disaggregation of the population on a three-dimensional building model (cf. for methodological details, see Sapena et al., 2022). Beyond, the building morphologies and patterns were used to assess the character of the neighborhood into precarious or non-precarious settlements (Kühnl et al., 2021) – a physical approach that is capable to proxy the social group of deprived populations to a certain degree (Wurm and

Taubenböck, 2018). Various characteristics such as small building sizes, low building heights, complex, dense, and organic patterns were used for classification of precarious settlements (see Kühnl et al., 2021; Taubenböck et al., 2018). Lastly, medium and high-resolution satellite data was used to monitor the development of Medellín and with it the increase of landslide exposure figures. Overall, more than 115,000 inhabitants (about 5%) were found to live in high-risk landslide hazard zones in Medellín. Interestingly, around 99,000 of them are precarious dwellers (Fig. 5). This reveals inequality for this social group

due to residential locations, as for the entire city only 40% of the population are classified precarious, while in landslide prone areas it is more than 80% (Kühnl et al., 2022). These findings are particularly relevant since we measured an annual growth of 150 built-up hectares per year since 2016, mainly taking place in precarious neighborhoods, which resulted in an increase of 1% of exposed areas in a period of only five years.

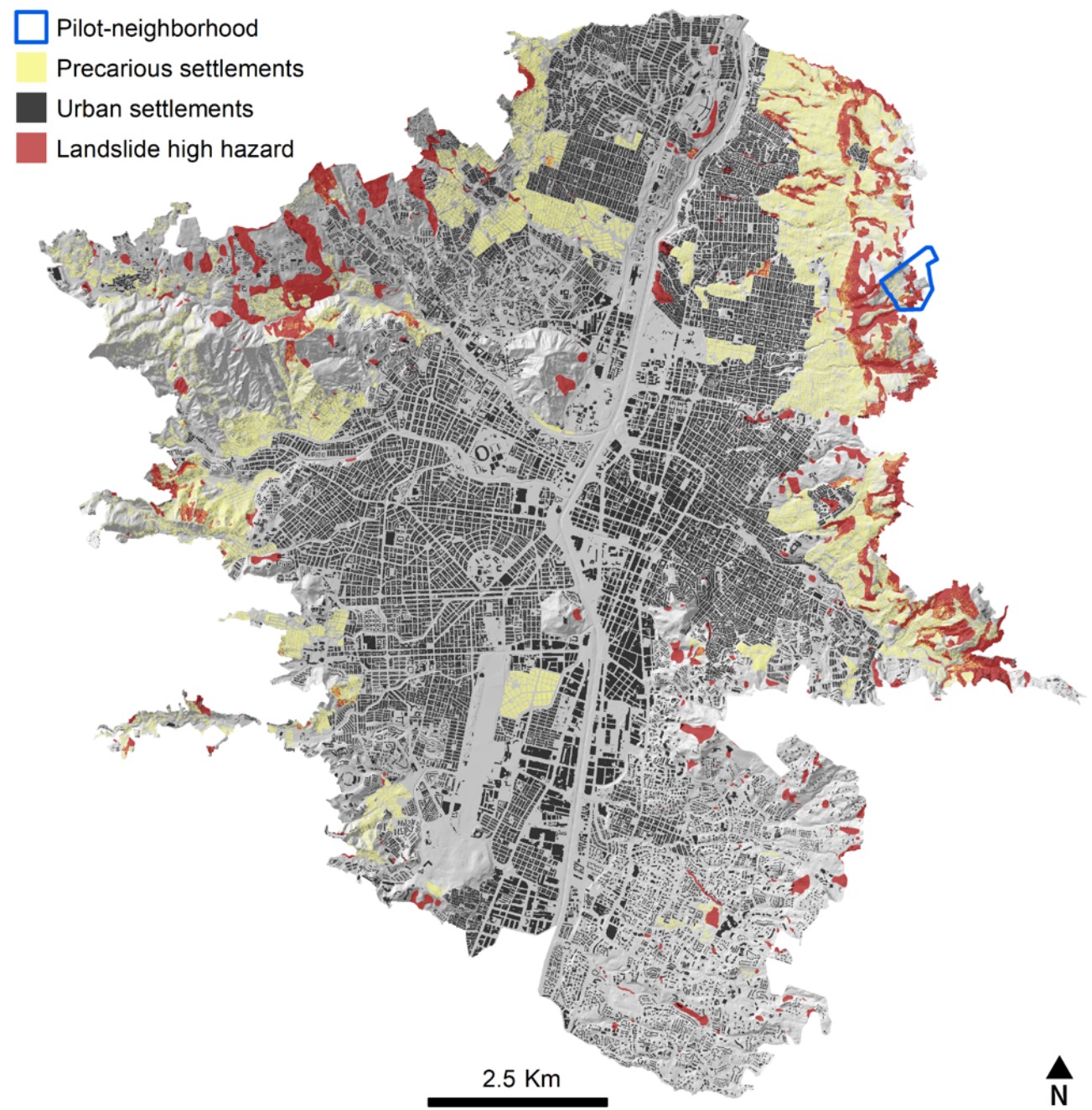

**Figure 5.** Exposure of built-up areas in the urban boundaries of Medellin with the location of precarious settlements (in yellow), as well as areas at high risk of landslides (in red). In the background, a hill shade map displays the topography. The data used are official cartography from the open data platform of the city of Medellín (Alcaldía de Medellín, 2023).

Using remote sensing methods, census, and official data sets, it thus becomes possible to locate landslide-exposed areas and quantify the exposed population and social groups. This large-scale approach allows for an approximation, but locally very specific conditions arise, for which higher resolutions of data are needed.

### 3.1.2 Landslide Hazard at Pilot-Neighborhood

The official data used in the city-wide analysis are highly relevant; however, their aggregated information often does not satisfy the local specifics, thus a higher level of spatial and thematic detail and accuracy is required. This can be achieved by field investigations. A detailed geological map of Bello Oriente and its vicinity has firstly been generated in 2020 and was re-evaluated in 2022. Additionally, several electrical resistivity tomography (ERT) profiles as well as four exploratory drillings were conducted to achieve a better understanding of the geological conditions. As Fig. 6 shows, a major part of the pilot-neighborhood is composed of colluvium material (green and turquoise colors), which is prone to landslide activity. The bedrock consists of the Medellín Dunite Formation. Parts of the rock mass which are weathered and not dislocated due to landslide processes can be named as Saprolite. According to earlier studies (Tobón-Hincapié et al., 2011) and as was confirmed by the geological investigations carried out in this project, the Medellín Dunite shows signs of so called pseudokarst processes. The dissolution processes taking place in the rock mass go back to weathering, generating a strongly indented relief of dunite top, caves and joints filled with clay material of up to 1.0 m width, which reach a depth of up to about 60 m (Rendón-Giraldo, 2020; Breuninger et al., 2021)

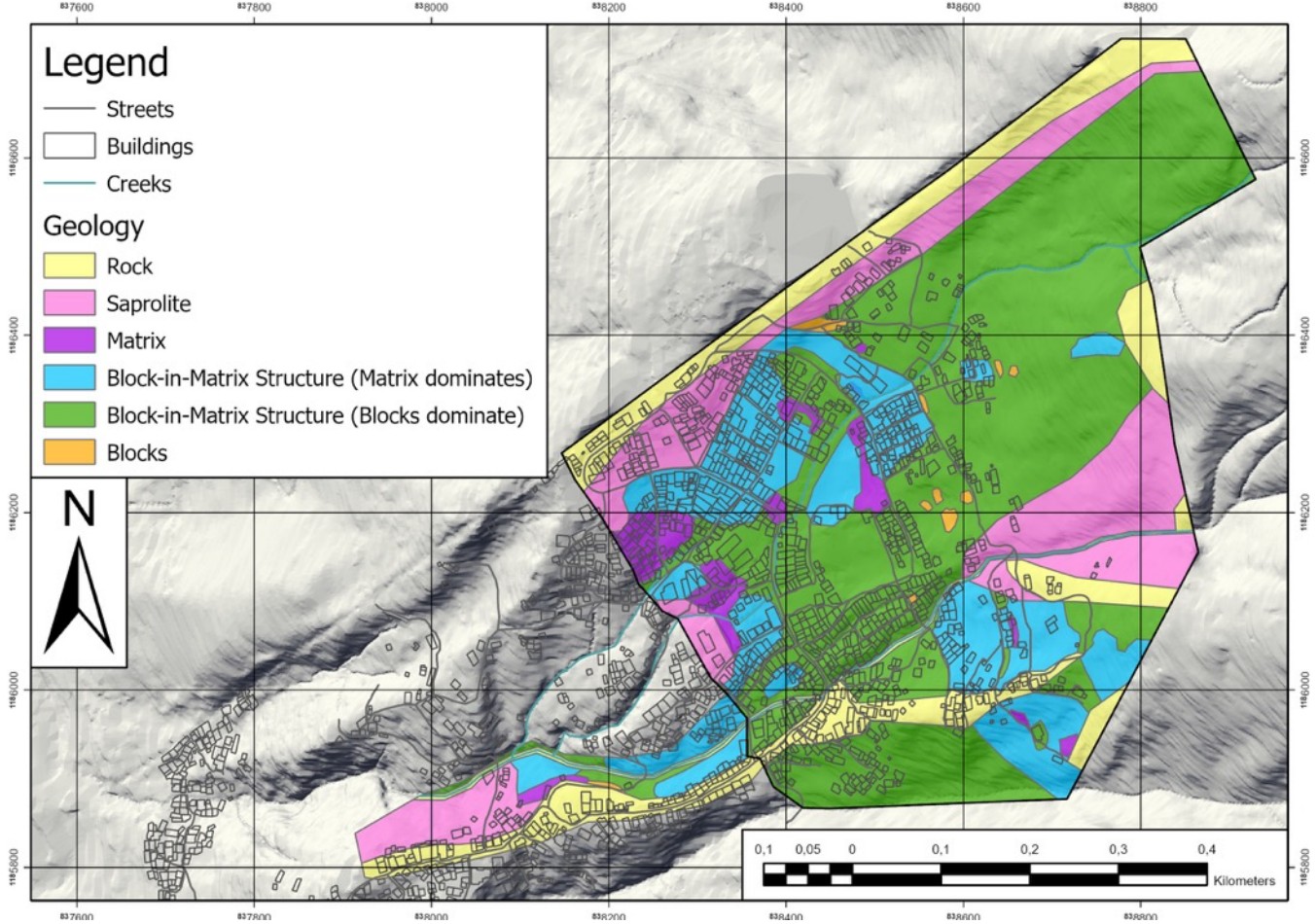


**Figure 6.** Geological Map of the pilot-neighborhood. The major part above the settlement of Bello Oriente is composed of colluvium material (green and turquoise colors), which is prone to landslides (Breuninger et al., 2021).

The combination of weathering processes and tectonics has a major influence on the landslide hazard in Bello Oriente. Figure 7 shows the tectonic setting of Bello Oriente and its near vicinity. The map was compiled by combining the results of hillshade

analyses and structural information collected during detailed field investigations and scanline analyses. The orange-colored joints (E-W strike) belong to a fault system which, based on the ERT profiles, seems to create a graben-structure in the region between the drillings B2 and A1. Due to this graben-structure, a slight increase in the colluvium thickness (up to 10 m) at the center of this structure can be recognized. The green-colored joint set (NNE-SSW strike) is oriented roughly perpendicular to the E-W striking fault set and runs nearly parallel to the slope. This joint set could promote landslides as it might act as a zone

of weakness in the rock mass, favoring water infiltration and possibly back scarps.

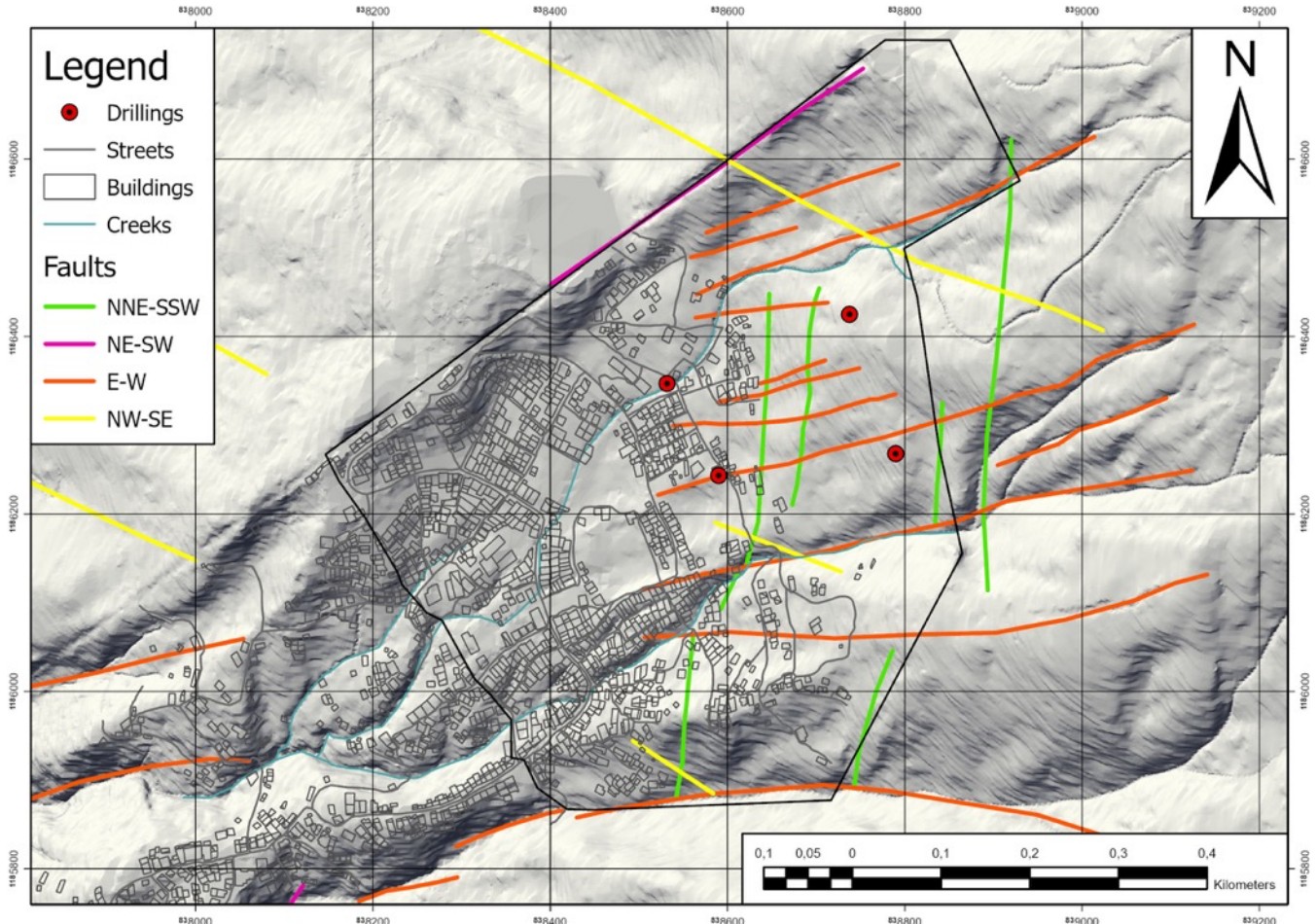

**Figure 7.** Tectonic map at the neighborhood scale of Bello Oriente. The E-W as well as the NNE-SSW striking fault systems are suggested to influence the rock mass mechanics and thus the landslide hazard in the current project area.

The most relevant landslide process has been identified as rotational slides of different sizes in the soil cover above the dunite. Thereby in general, the depth of the soil cover controls the size of the probable landslide events. As rainfall is the most important trigger for landslides in the region, the hydrogeological conditions are key to understand where future landslides are likely to occur. Due to the high heterogeneity of the subsurface it however is impossible to gain detailed local knowledge of the subsurface water flow.

The hazard map for Bello Oriente was established by combining the information of the geological investigation with mapped landslide processes on-site and the recognized events of the SIMMA database, which contains landslide events in the Aburrá Valley, classified according to their type of process, their location, and the event date. The analysis showed, that the frequency - magnitude distribution of landslides in the region of Bello Oriente can be characterized by three scenarios:

- small landslides (annuality of 30 years), an area of up to 4000 m$^2$ and a depth of 5-10 m,

- medium landslides (annuality of 100 years) with areas of 4000-15000 m$^2$ and depths of 20-30 m and

- large landslides (annuality of 300 years or more) with areas of more than 100000 m$^2$ and depths of up to 30 m.

The areas, where these scenarios might occur within the project area were determined based on the existing landslide phenomena, information of the soil cover thickness and a critical slope angle for landslides (FOEN, 2016). The latter was determined by 2D limit equilibrium analyses using the code SLIDE-2D (Rocscience) and shear tests on soil samples. Both indicate that the critical slope angle for landslides varies between 22° and 24° depending on the water saturation. Using the "Fahrböschung" approach (Heim, 1932; Evans & Hungr, 1993) with an approximated worst-case runout angle of 20 °, the potential runout of the landslides was estimated.

Figure 8 shows the hazard map for the most frequent 30-year event. The map contains not only the information about the hazard and the run-out area, but also the location and dimensions of the mapped landslides in Bello Oriente.

The local-level analysis revealed a high vulnerability in Bello Oriente, particularly in the LEWS pilot-neighborhood (38ha), where 49% of the roughly 4,600 residents (in 2021) live in an area with a "high" landslide hazard, 23 % in the "residual hazard" area and 28 % in an area, where up to now no landslide risk could be detected (Table 1)

| Landslide Hazard | Number of buildings | Population (%) |
|---|---|---|
| category "high" | 680 | 49 |
| category "residual" | 258 | 23 |
| no hazard/risk | 347 | 28 |

**Table 1: Statistics summary of the vulnerability analysis for the LEWS pilot-neighborhood including the exposed buildings and the population in %.**

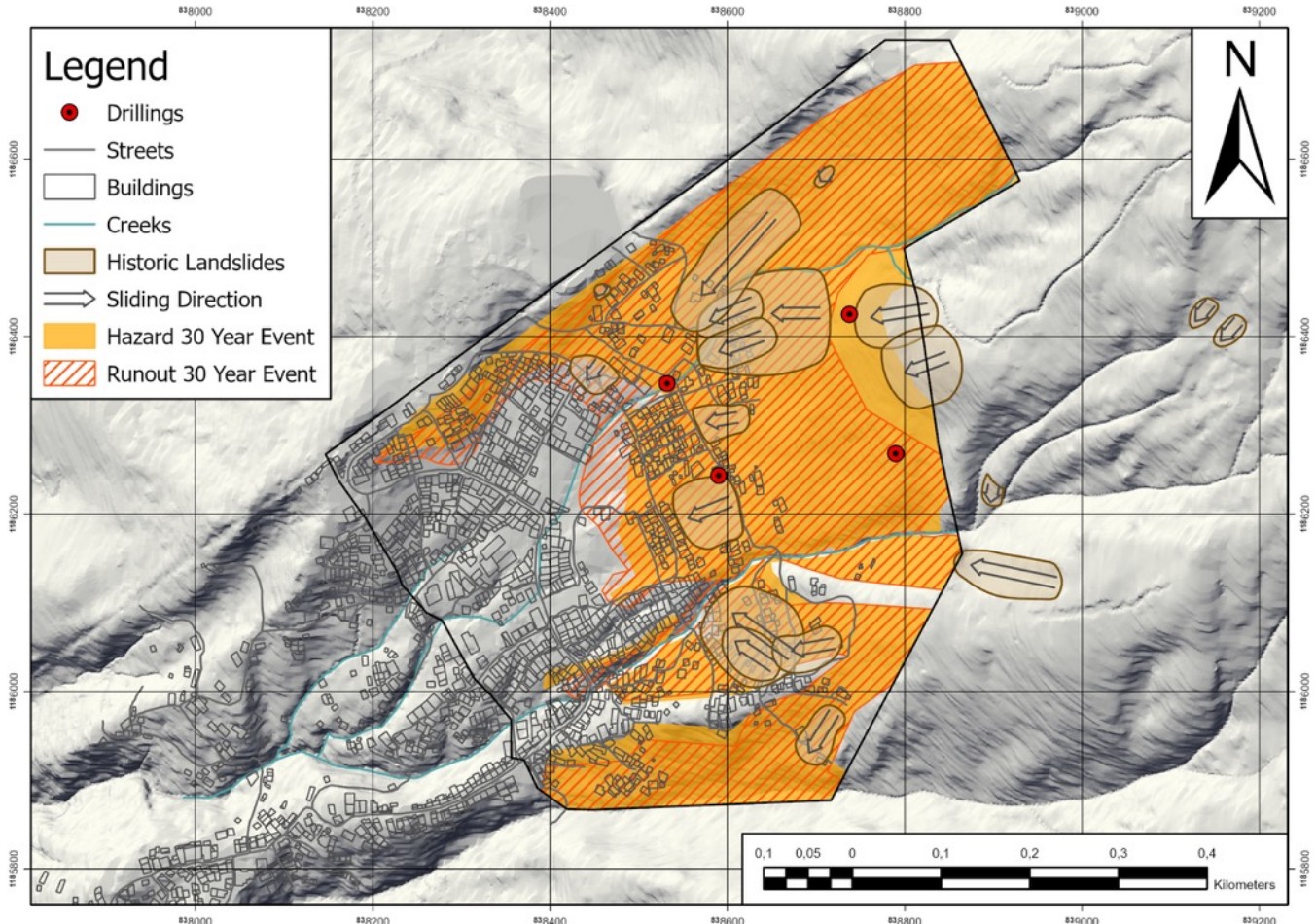

**Figure 8.** Hazard map (30-year event) of Bello Oriente and its vicinity at the neighborhood scale. The orange area marks the hazard area, where scarps could occur, the red shaded area represents the potential runout areas. The grey shaded overlays represent landslides, which were mapped in the field and are suggested to belong to the 30-year event-category.

It was estimated that 72% of the population of the pilot-neighborhood are exposed to landslide hazard. This finding roughly correlates with the hazard perception of the residents of Bello Oriente. In a representative survey in 2020 (259 participants) over 60% of the interviewees rated the likelihood of a landslide nearby their home as high to very high and 65% of participants were very concerned about it.

### 3.2 Landslide Monitoring

While the results of the detailed landslide hazard and risk analysis characterizes the causes for likelihood of initiation and the effect of a landslide at any point in the area of the test-site, they do not allow to make a prediction where and when the next landslide will occur. To achieve this, the slope needs to be continuously monitored. Two complementary monitoring concepts were pursued in the living lab approach: automatic sensor-based monitoring and manual community monitoring.

### 3.2.1 Automatic Sensor-Based Monitoring

The general concept of the sensor-based monitoring system is to predict the future behavior of the observed landslide prone area. This is achieved based on detailed hydrogeological and geotechnical models, which have been calibrated by observational data from hydrogeological field tests, geotechnical laboratory tests and a dense low-cost geosensor network. By including the triggering process in the models (e.g. intense precipitation leading to high ground water levels), it is feasible to issue first general notifications several days to hours in advance of a critical phase concerning the stability of the slope. When the onset

of slow, but increasing movement is detected, spatially precise early warnings can be issued. In case of further or sudden strong acceleration of the slope, (evacuation-)alarms can be disseminated at least hours to minutes prior to a catastrophic event, allowing people to leave the endangered area in time.

In order to reliably detect the initiation of landslides – especially in the "small" scenario within a 30-years repeat period

described above (rotational slides, usually 10–100 m wide) – without previous knowledge of the exact location, area-wide spatially and temporally highly resolved and accurate deformation observations are required. To achieve this high measurement density at an affordable cost, a new geosensor network consisting of a combination of horizontally installed Continuous Shear Monitor (CSM) (Singer et al., 2009) and wire-extensometer measurement systems combined with low-cost wireless sensor nodes (*Inform@Risk* Measurement Nodes) based on Internet of Things technology has been developed and implemented in

Bello Oriente (check Fig. A1, Gamperl et al., 2021; Gamperl et al., 2023). While the CSM and wire-extensometer systems provide continuous, spatially highly resolved deformation observations along measurement profiles, the wireless sensor nodes add punctual observations of deformation and triggering mechanisms based on the integrated MEMS sensors and other external geotechnical and hydrological sensors (e.g. piezometers).

The Measurement Nodes (Fig. 9a and b) are based on a newly developed sensor PCB and the Arduino MKR1310 microcontroller, which provides a LoRa® (Long Range) radio transmission link. The new LoRa® wireless technology allows the transmission of small data packets across large distances of several kilometers while only requiring very little power. This makes the development of sensor networks possible, which can be distributed in a wide area with little additional infrastructure required. The nodes can be operated with six standard AA batteries for a very long period (up to about 2 years) and by using

small solar panels continuous operation is possible. By using affordable hardware and materials and utilizing simple manufacturing and installation techniques (e.g. 3D printing), the cost of the complete system is comparably low.

While the Measurement Node already has a suite of MEMS sensors on-board (high-quality tilt sensor, thermometer and barometer), additional analog and digital sensors can be connected via the onboard 24 bit A/D converter and several digital

interfaces.

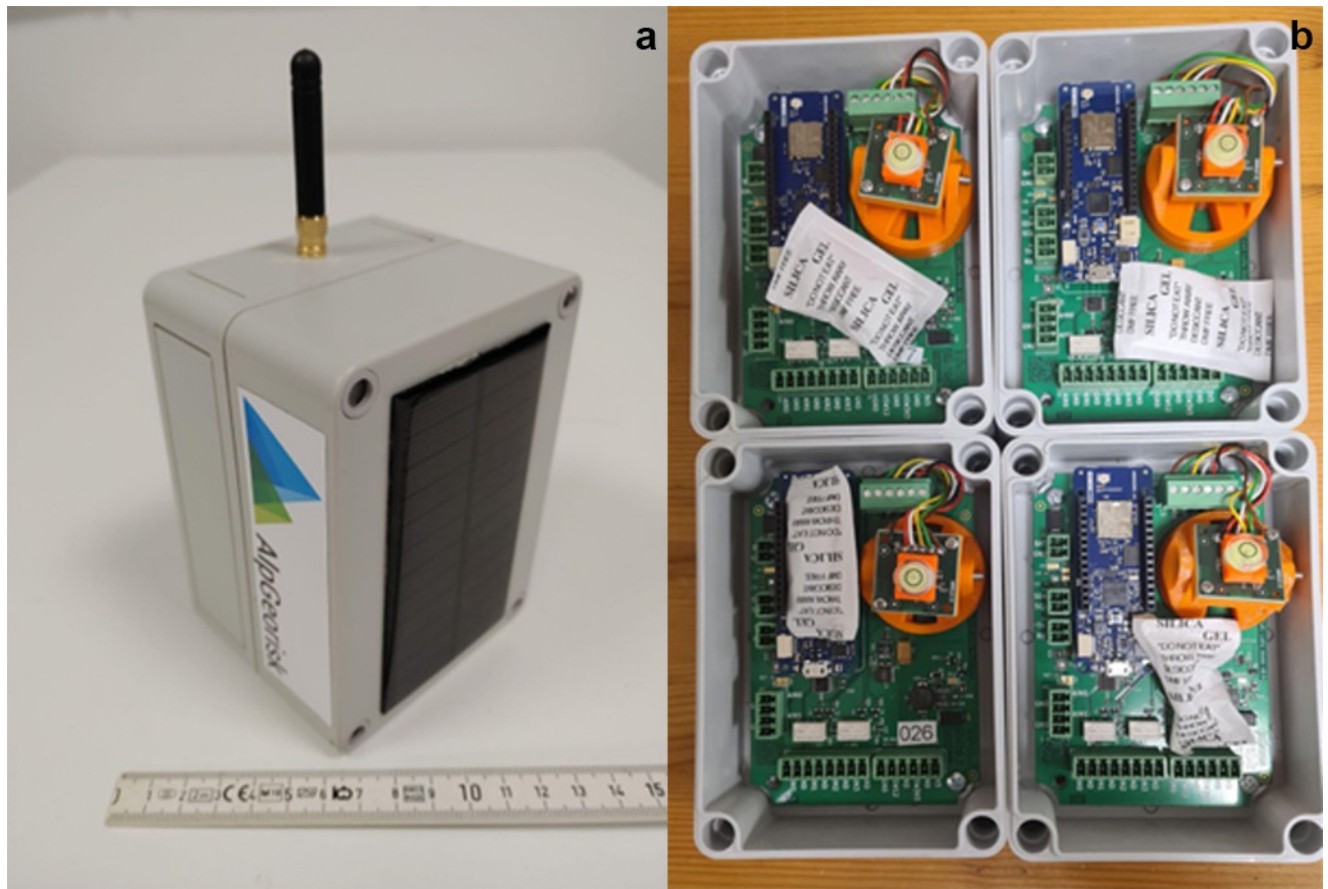

**Figure 9.** (a) Sensor Node with mounted solar panel for autarkic operation (b) Inside view on four sensor nodes. The Arduino microprocessor MKR WAN 1310 (blue PCB) is connected to the Inform@Risk sensor PCB (large green PCB), which contains
a suite of different sensors including a thermometer, barometer, 4-channel 24 bit ADC and an accurate tiltmeter, which is mounted on an orange rotatable base to make leveling of the sensor independent of the box orientation possible.

One example of an external sensor is the Subsurface Probe (Fig.10a and b), which consists of a combination of one or more

inclination sensors and a water pressure sensor. The subsurface probe is installed into a small diameter borehole (1 1/4 inch)

and can function either as a chain inclinometer system (multiple inclination sensors) or as a simple motion detection system

(one inclination sensor). The water pressure sensor gives information on the ground water conditions at installation depth.

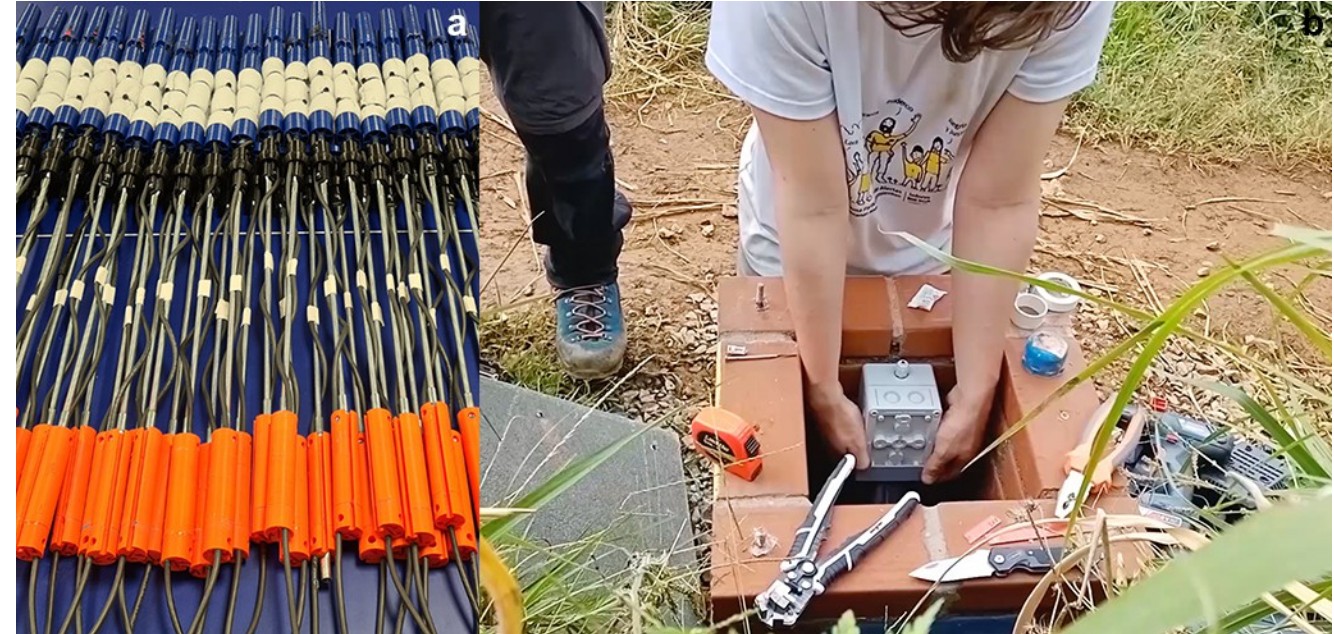

**Figure 10.** (a) Sensor Probes ready for installation. The piezometer (blue) is located at the bottom of the probe. Above follows a ball joint (black) which exactly fits into the used installation pipes and functions as reference points for the inclination measurements. Above follows the first inclination sensor (orange), which - when installed - is followed by another ball joint (2nd reference point located exactly 1 m above first joint). If required, further inclination sensors can be added. (b) installation of subsurface probe into a brick seating cube.

The Measurement Node and the Subsurface Probe have been developed as open source. Details on the sensor system can be found on the Inform@Risk Wiki page (www.informatrisk.com).

In total 115 measurement nodes, of which 45 have a subsurface probe attached to them, as well as about 1 km of CSM/EXT measurement lines have been installed in the project area (check Fig. A2). The sensor system is operational since fall 2022 and has been in operation without major complications.

The acquired data is relayed to a data cloud via three data gateways which have been installed in a redundant configuration in the pilot-neighborhood. These via internet connected gateways receive the LoRa® transmissions of the measurement nodes and the data of the CSM/EXT system on site and transmit the data to the data cloud. In the living lab approach the AlpGeorisk ONLINE (www.alpgeorisk.com) data management platform is used to manage the large amounts of data (several GB per month) which are generated by the geosensor network. This platform is responsible for data quality management, calibration, storage, analysis and visualization via web-portal as well as system management and control and notifications.

In order to achieve an optimal effectiveness of the system in terms of risk reduction, the density of the observations was varied throughout the project area based on the results of the risk assessment. In less risk prone areas, the sensor density is significantly reduced. While this approach ensures an optimal cost-benefit ratio of the entire system, it must be emphasized that especially

small low intensity and very unlikely events might not be detected. To fill these gaps, the sensor-based monitoring was complemented with people-based monitoring.

### 3.2.2 Manual and People-Based Monitoring

The inhabitants of Bello Oriente have good empiric knowledge of the different landslide signs and conditioning factors. For example, before a landslide in 2017 that was caused by a septic tank with poor maintenance, residents detected changes in the slope such as cracks, water ponds, new bumps in the road, tilted trees, among others. After monitoring the evolution of those signs over the course of a week, people of the community alerted the neighbors who were able to evacuate just in time.

Therefore, it is a fact that the inhabitants of the high hazard areas, especially those who have lived there the longest, perform visual monitoring of their surroundings. The community also looks for water pipe leaking, which is one of the main anthropic landslide triggers, together with informal house, road and path construction.

The importance of visual monitoring was discussed and preliminary experiences were exchanged in numerous workshops with the community. Moreover, some manual instruments with recycled materials were constructed (Fig. 11b shows a simple inclinometer) in order to show that it is possible to manually monitor and measure deformations (UNGRD, 2013) equivalent to the functionality of automatic sensors.

In deviation of the international standard ISO 22327 (2018), community leaders of the pilot-neighborhood opted against the creation of a specific landslide monitoring group, and preferred a decentralized integration of monitoring activities in existing community groups.

### 3.2.3 Socio-spatial Integration of Monitoring System

In order to be accepted, any spatial intervention in informal settlements has to be developed with residents (Imparato et al., 2003; Werthmann, 2021). This is especially true for Bello Oriente, as many of the 4,600 residents display a fundamental distrust against the government due to feared resettlement measures. A main objective of the socio-spatial integration of the technical elements of the LEWS was to serve as a natural reminder of landslide risk, while increasing the quality of public spaces in their daily use, the acceptance of the technical system and decrease vandalism (goal 5). Accordingly, particular attention was paid to involve the residents right from the beginning and develop trust through transparent communication, while gradually building up risk awareness and monitoring skills by joint activities on site. From 2019 to 2022 over 40 workshops and community meetings with over 1,000 participants and 19 organizations (NGO', CBO's) were held (an example for one typical workshop setting can be seen in Fig. 11a). Different types of designs for the sensor locations were developed, ranging from simple sensor covers and markers in remote areas to painted cube seats and benches with information boards in the public spaces, which can be used as meeting places and recreational spaces. Tree plantings and micro gardens in the form of small flower beds were co-created to support the self-proclaimed goal of the pilot-neighborhood to transform into an 'eco-barrio'.

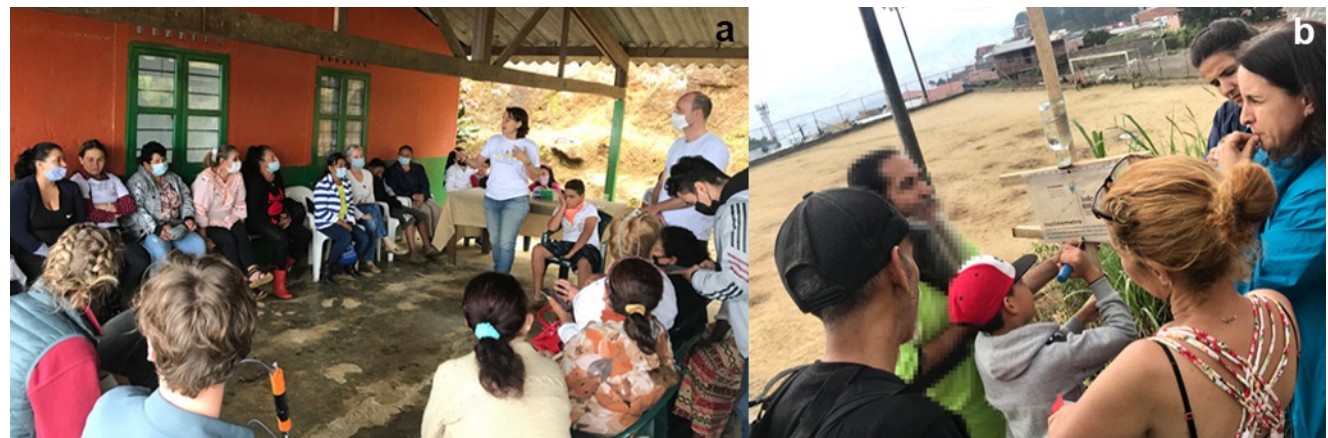

**Figure 11.** (a) Typical community workshop setting and (b) manual monitoring installation of a simple inclination meter by residents.

Prototypes of cubes and benches were tested on site with different materials which are weather resistant and have low susceptibility to vandalism and theft. Finally, because of its low cost, local availability, and its familiarity to local craftsmen, brick was chosen (Fig.12a, b). In addition, an overall signage and information system was developed to explain landslide risk and the LEWS (Fig. 12c). Continuous information campaigns were organized using a graphic language with comic characters that provide easy to understand information and allow for emotional attachment (Fig. 12d).

Sensor locations were discussed and continuously adjusted in joint field walks with the residents. Although the sites are located in public spaces, the permission of landowners and neighboring residents was considered of importance to gain acceptance and avoid future conflicts. Thereby, the local expertise on the territory was essential to cope with complicated landownership and criminal activities of gangs controlling parts of the informal settlement.

Before the construction works were carried out by professional firms, tree plantings were organized with the community to mark the sensor locations. Moreover, a concept of sensor 'godparents' was established, where nearby residents agreed to care for the sites. Some bench locations were refused by the residents, being concerned that members of criminal groups would use them as meeting places. Finally, 15 cube seats, six brick benches, nine information boards, and two murals painted by local artists (Fig. 12e) were installed in the upper part of the pilot-neighborhood by mid 2022 (Fig. A3 shows all locations in plan).

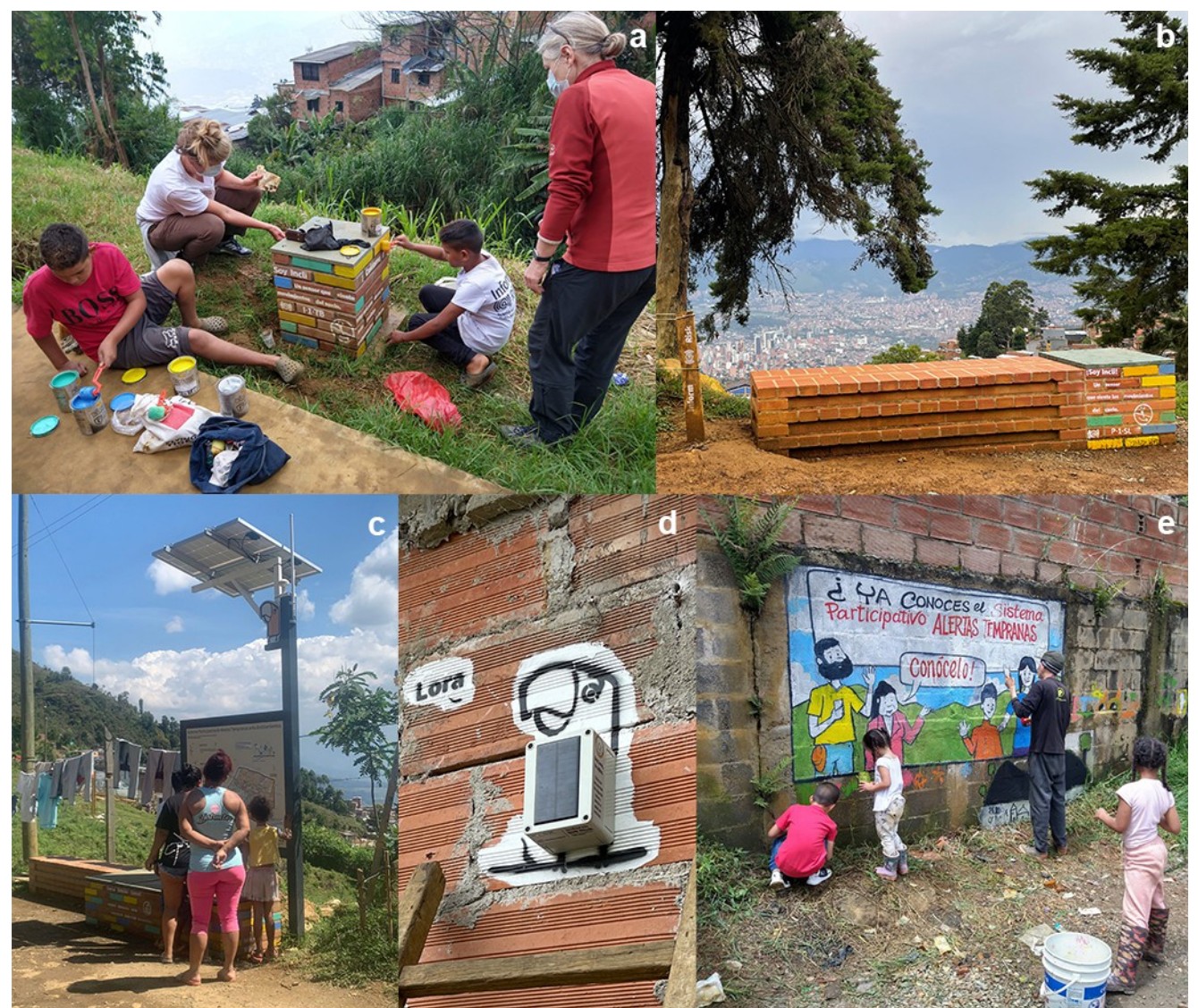

**Figure 12.** (a) Collaborative painting of a brick seating cube with a subsurface probe inside (b) a micro garden next to a brick bench sheltering a Subsurface Probe (c) Brick bench housing the central monitoring station and an information board (d) Inclination sensor on a private home with the Inform@Risk mascot "Lora" (e) murals painted by local artists with the help of children.

A representative survey (253 participants) for the whole pilot-neighborhood and qualitative interviews of residents (in total 28) living nearby the enclosures have been conducted shortly after installation. It was found that the degree of socio-spatial integration of the LEWS varies. In the survey, 50% of the residents were not even aware of the enclosures. This can be partly explained by the fact that the installations are located in the upper half of the pilot-neighborhood and that the survey was done shortly after completion of installation. 28% liked the new elements while only 17% of the residents knew about their

monitoring function. The interviewed residents living near the enclosures generally liked them, but many also displayed only rudimentary knowledge about their function.

A different indicator of socio-spatial integration is the degree of vandalism. Here, also a mixed picture emerges. After six months of use, most of the communal tree plantings were cut. In contrast, the micro-gardens are actively tended to and the cubes and benches reveal only minor traces of damage (mostly caused by children).

## 3.3 Warning

### 3.3.1 Warning Levels

In 2023, the LEWS will still be in a "learning phase", where the sensor data, on site observations, and numerical modeling results are used to define the warning thresholds for the different individual sensors and sensor combinations.

In order to assess the short- to medium-term hazard level, mainly the triggering factors rainfall and groundwater height are considered. The medium-term hazard level (pre-warning) is about one week to one month and is based on previous accumulated rainfall (two weeks / four weeks), current groundwater levels and the weather prediction and long-term seasonal climate data. The short-term hazard level is based on current data of the sensor system concerning groundwater and deformation information. This is the basis for issuing warnings concerning the situation in the next hours to days.

To determine the according thresholds on the one hand, as soon as enough data has been acquired, time series analyses will be performed to identify causal and temporal relationships between short-, medium- and long-term rainfall and groundwater levels. On the other hand, the hazard level is determined using groundwater level thresholds at e.g. 50, 75 and 90 percent of the critical water table derived from the geological/geomechanical models created during the hazard analysis. The threshold values thereby are determined individually for different geological homogeneous zones throughout the pilot-neighborhood and are applied to the according sensors in this area.

Four basic warning levels have been developed in collaboration with the Medellín risk management agency DAGRD and the civil society organizations, combining both qualitative and quantitative thresholds, as well as the responsibilities and expected detailed actions for each level. In table 2 a simplified version is presented, but a detailed version with further instructions was distributed to the community. As an augmentation to the three proposed warning levels of the international standard ISO 22327 (2018), a first level (green) has been added, in order to promote constant monitoring and to avoid increasing risk activities such as incorrect construction practices and leaking water pipes. The second level (yellow) promotes more detailed monitoring, when any changes in weather conditions and/or soil movement are detected. The third level (orange) promotes to prepare for evacuation. The fourth level (red) means immediate evacuation. The third and fourth level are critical warning levels that are only reached when significant deformation is detected.

Based on how many and which neighboring sensor nodes show deformation, the affected area and landslide mass are estimated and reported to the risk management authorities. If a further or sudden strong acceleration is detected, it is planned that the system can issue an immediate (evacuation)-alarm using acoustic signals.


Depending on the exposed population, deformation rate and the affected area, different actors (experts, trained community members, first responders, whole population) are informed. Usually, warnings will be checked by an expert from Medellín's risk management agency DAGRD before they are sent to the inhabitants. Only when at least two neighboring sensor nodes show strong acceleration in similar direction and at the same time it is planned that warnings are issued without review.


| Alert level | Action |
|---|---|
| **Stable - normal**<br>No changes in the slope connected to landslides | **Prevention** |
| **Caution**<br>There are one or more signs and/or changes in the slope and weather that could be connected to landslides | **Observation** |
| **Alert**<br>There are several signs and/or changes in the slope and weather that could be connected to landslides so be prepared to evacuate | **Preparation** |
| **Alarm**<br>The signs and/or changes in the slope and weather indicate that a landslide is imminent, evacuate immediately. | **Evacuation** |

**Table 2.** Simplified warning levels.

### 3.3.2 Warning Dissemination Channels

Several channels are used to assure that the warnings reach the people at risk. First, direct communication via a WhatsApp group with members of the community and representatives of governmental risk management organizations has been installed. Photos, written and audio messages are shared in the group to describe the situation and alert in case of a possible emergency. Second, in the framework of the project two instruments were installed to send alert messages and alarm sounds when needed. They consist in one sound system with eight speakers and one siren, both with solar panels to avoid disruption in case of failure in the cable energy grid (Fig.13a). The speaker system is used not only to sound an alarm, but also to send a predefined message


that contains the name of the sector of the potential landslide together with a basic description of the actions, depending on the warning level. At the time of writing, the speaker system is operated by community members, who were trained in the use of the instruments. In the near future it is expected to install a control to be able to activate the system remotely by the risk management authorities.

It is planned to use a smartphone application as a third alert channel. The so-called Inform@Risk App was developed to serve as a hub of both general and immediate information concerning the current state of landslide risk. It can display a range of relevant information for inhabitants of the study area like the currently active warning level as defined by the LEWS, status and data of individual sensors in the project area and pre-defined meeting points in case of alert. In case of emergency, users of the app can receive push notifications and detailed information about spatial constraints of the ongoing event. The app also

encourages users to participate in the warning dissemination and decision-making process by enabling them to upload reports containing photos and descriptions of local developments (e.g. new sinkholes or cracks in infrastructure) to notify authorities directly (Fig. A4).

Additional challenges for the development of the smart phone app were added due to the location of the pilot-neighborhood in an informal settlement. Smart phone use is limited and most inhabitants do not have access to reliable internet; therefore, there

is greater need than usual for the app's functionality to be available offline. Smart phones in use have limited memory space and processing power, imposing challenges for interactive content like large dynamic maps. In addition, inhabitants are worried about tracking through personal data. The app has been tested by a small number of residents and is currently on hold until it is clear if the LEWS can be operational for the municipality of Medellín.

### 3.3.3 Warning Network

The landslide that occurred in 2017 is strongly present in the memory of the inhabitants of Bello Oriente. Having caused no casualties due to the timely alert disseminated by the community itself and an evacuation just seconds before the landslide was triggered, this event serves as a baseline for future landslides. However, future events might not show preliminary signs and might happen at night, when it is more difficult to visually identify the movement, therefore increasing the risk of having casualties. In order to disseminate the warnings, a "warning network" was preferred by the governing authority and the

residents in favor of a "warning chain", where the failure of one link can disrupt the whole warning chain. Considering that the population is large and highly dynamic, with constant new incomers in the pilot-neighborhood, the residents of Bello Oriente decided not to create specific warning groups but to promote for every neighbor to be responsible to disseminate the warning among the closest neighbors. In order to achieve redundancy, other alert methods were suggested as using WhatsApp groups, the sound and siren system, the mobile App (in the future), and also to involve the Local Community

Council and local civil society organizations in the alert dissemination. The effectiveness of this alert network has to be evaluated through regular emergency drills in the future.

As part of the communication strategy the project installed several signs to explain that every person is responsible to be aware of the landslide risk, to be prepared and to evacuate in time. This was done, not only acknowledging the fact that the risk management authority and emergency services will take a long time to reach the neighborhood, but also follows one of the main premises of the Colombian Risk Management Law.

## 3.4 Response Capacity

According to the risk management Colombian Law 1523 of 2012, risk management is the responsibility of every inhabitant of Colombia.  Before this law, all the responsibility resided with the government. Following this concept, the research project has been working on strengthening the risk management capacities of the community and the local organizations, regarding risk knowledge, basic risk reduction and mitigation measures, as well as response capacity. While the expected increase in monitoring and dissemination capacities of the LEWS prototype should lead to more timely responses, response capacities still have to include to the possibility of a fast-onset landslide with just minutes of warning time due to seismic activity or extreme weather events.

### 3.4.1 Evacuation Planning

In Bello Oriente, the first respondents are the community members themselves, which is fairly common for informal settlements in fringe areas (Heffer and ELLA Brazil, 2013). Bello Oriente, which is located in the upper hills of Medellín, can only be reached through steep and heavily trafficked narrow streets. It is almost impossible for the local authorities and the emergency response teams of the nearest firefighter station to swiftly reach the area in order to lead a rapid evacuation (in one emergency drill, it took the firefighters 40 minutes to arrive). Therefore, the training of residents for a rapid evacuation scenario is crucial. Accordingly, several community workshops have been held to identify best evacuation routes and meeting points. The evacuation strategy was designed for different scenarios of day/night time and weather conditions. Basic actions were established for each warning level together with the local community, the risk management authorities and the civil society organizations. The participants of the workshops were asked to share the gained knowledge with the other inhabitants of the pilot-neighborhood.

### 3.4.2 Evacuation Training

The first two community landslide evacuation drills ever held in Medellín have been performed in 2022 with about 100 participants each in Bello Oriente (Fig. 13b). They were developed with the support of the local risk management authority, firefighters and volunteers from different organizations, together with volunteers of the community. In order to execute the drill, the area was divided into several sub-sectors, where the evacuation was led by the volunteers, who were promoting the evacuation door to door. Afterwards, participants evaluated the drills and identified various improvements. For example, residents with permanent and temporary disabilities need to be identified in order to assign neighbors, who could assist them

in case of an emergency; or additional sound systems and megaphones are needed to reach all residents. Overall, it became clear that the participation of civil society organizations needs to be strengthened and the numbers of participating residents increased. The risk management authorities need to continue regular emergency trainings and biannual drills in order to establish a familiar routine for the residents at risk. As these drills are a novelty for the risk management authorities, proper protocols need to be further developed.

Due to geo-physical and governmental constraints, one of the still unsolved challenges is the best timing of an evacuation. Landslides in the pilot-neighborhood can be either sudden or slow on-set. Therefore, a mass movement might take hours, days, weeks or months, to completely move after first signs have been detected. In addition, anthropic landslide activation plays a major role in Medellín (Alcántara-Ayala and Oliver-Smith, 2019). A landslide with a naturally slow behavior might be suddenly accelerated by human activity (leaking of pipes or septic tanks, slope cutting, insufficient maintenance of drainage channels, etc.) rendering warning times for evacuation extremely short. Accordingly, the research project developed multiple workshops for residents to increase the awareness of correct water management and construction practices, as well as the importance to be prepared to evacuate with very short notice. It is hoped that in the future the training of geo-sensors during and beyond the test phase will precise the quantitative thresholds of triggering factors of mass movements. Thus, the issuance of an evacuation order might become more accurate in relation to landslide activity.

Beyond the best timing of evacuation, another serious challenge of evacuation lies in the lack of proper housing replacement options for evacuees. Once an evacuation order is issued, it is unclear for evacuees, when and if they ever can return to their homes. Residents in informal settlements rarely have private options (neighbors or friends) to evacuate to for longer time periods. Public agencies are not allowed to build temporary shelters before a landslide has actually occurred. Rent subsidies for evacuees are complex to obtain, can take months to process, are short-lived (three to six months) and are in general too scant for low-income families to afford a new location (Mesa Interbarrial de desconectados, 2011; Rojas et al. 2022). This means that many families are currently fearing an evacuation order and are reluctant to leave their homes, once an evacuation order has been issued. Better housing replacement options need to be developed by the government not only in order to increase compliance with evacuation orders, but also to assure resident's acceptance of an early warning system that they eventually might misperceive as a covert governmental tool for eviction.

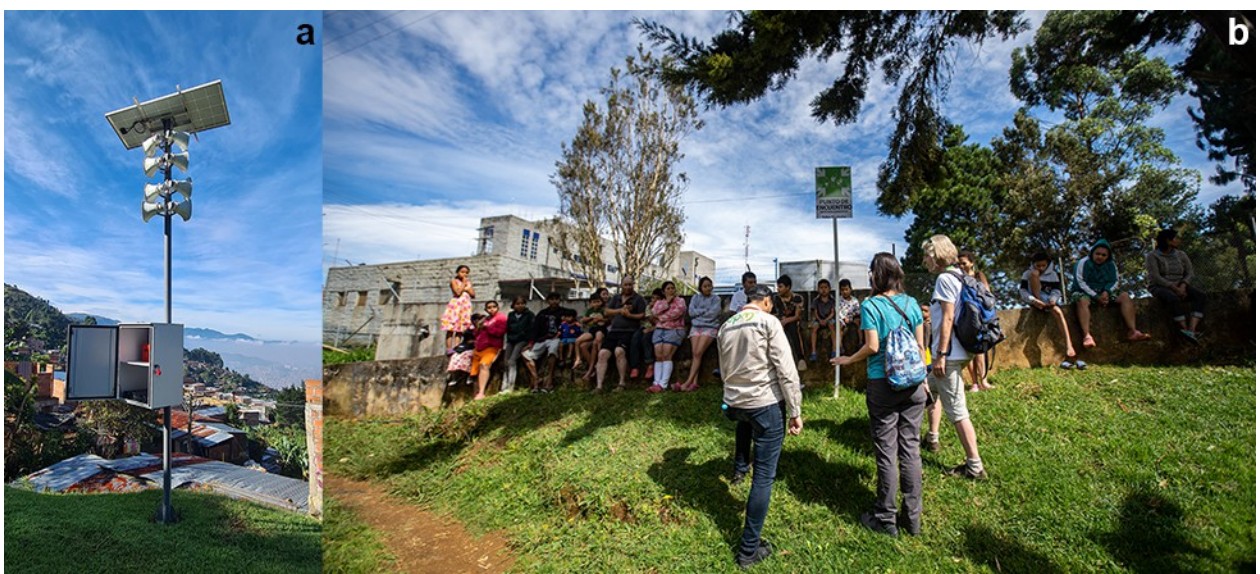

**Figure 13.** (a) siren system that can be manually operated and (b) residents congregate at one of the meeting points after an emergency drill.

## 4 Living Lab Experiences and Conclusions

The development of a LEWS is a socio-technical challenge. The living lab approach revealed one possible pathway to integrate the multiple objectives of stakeholders, as well as the technical, scientific, social and planning expertise in such a way that the risk of a precarious settlement to landslides could be addressed. It is precisely the integration of such diverse issues as scientific concept, data and results, technical implementation, social acceptance, and political will that has to align. As the gained knowledge of living labs is generally context specific and not universally transferable to other cases (Schneidewind et al., 2014), we will critically reflect on the findings in the specific context of the study and try to deduct useful patterns for the development of LEWS in informal urbanization.

In the presented living lab approach, seven goals for an integrated LEWS against landslides in informal settlements have been formulated at the outset. The LEWS has to be 1) precise, 2) affordable, 3) socially integrated, 4) multi-sectoral, 5) socio-spatially integrated, 6) multi-scalar and 7) replicable. In the course of the research an eighth goal was found to be of high relevance and was thus added: 8) redundant. A prototype of a LEWS has been developed and handed over to the responsible risk management agency of Medellín (DAGRD), at the end of 2022. This prototype is not yet a fully functioning warning system; it has entered a test and calibration phase, and as previously stated the LEWS continuation is currently uncertain. It needs to be further refined and integrated into the disaster prevention procedures of the city of Medellín in order to get a firm understanding of its usefulness and potential replicability. The following conclusions have been drawn from the four-year research and development phase with the addition of some cautious prognostic statements:

1) precise

The automatic sensor-based monitoring system delivers temporally and spatially very dense data on deformation and trigger process activity. Based on the collected data, we believe that it will be possible to generate landslide activity warnings with to date unparalleled precision in terms of location and level of hazard. As even small movements in the range of millimeters can be detected by the sensor system, landslides can be identified and monitored at a very early stage of development. This allows

to detect critical changes in the landslide activity early on giving more time to react appropriately. Based on the experiences made from past landslides in the region it is believed that an early warning (without certainty of critical failure) can be issued several days in advance, an alert (with near certainty of critical failure) hours in advance to the event. Currently, the warning generation procedure and response is in development by the responsible risk management agency DAGRD and still has to prove its capability in the coming months and years. It also has to be noted that the automatic sensor-based monitoring system

can despite increased accuracies only sense deformation, where a sensor is actually located. It is designed for medium to large landslides with potential catastrophic impact. Smaller landslides (smaller than about 30m) cannot be detected and community-based monitoring remains essential (also in case of technical failure).

2) affordable

The prototype of the LEWS aimed to develop state of the art technology with the lowest cost possible in order to be generally affordable and thus be easily replicable. Unfortunately, a conclusive understanding of cost and affordability cannot be given yet as the further development and operation of the prototype is on hold due to legal issues by the municipality of Medellín and too many unknowns remain.

The costs expanded for the LEWS prototype by the German research team are not representative for the costs of future implementations as they include development costs, German administrative costs and German salaries. It also is not possible to assess the costs for the hazard and risk analyzes and social work in a generalized way as they greatly depend on the complexity of the situation in the individual project site and the level of preexisting know-how and administrative structures, which even within a single municipality can vary greatly. The development of the LEWS prototype required a large amount

of coordination with various stakeholders (200 meetings) and intense community engagement (40 workshops). A mature and replicable LEWS might require less coordination meetings in the future and community outreach might decrease. But this will depend on the structure of the community at-risk and the capacities of the risk management authorities in charge. Also, the cost for monitoring the sensor system day and night will vary considerably, if the system is added onto an already existing citywide multi-hazard sensor-monitoring system (like SIATA in Medellín) or if a stand-alone observation unit just for

landslides has to be funded.

The only component of the LEWS, where sound cost estimates are possible based on our current experience is the installation of the sensor-based monitoring system. Sapena et al. (2023) have performed a study on the cost for implementing the wireless geo-sensor network in different high-risk areas in Medellín which results in costs between € 5 and € 41 per inhabitant. The cost variation thereby is mainly related to differences in the population density and the required monitoring precision which was varied based on the level of landslide risk in the area.

In the end, only future long-term experience will allow to give sound numbers concerning the overall costs of implementing and maintaining the prototype LEWS. However, it is already obvious that the developed prototype with its increased efforts for risk analysis, monitoring and social work requires considerable budget and trained personnel for installation, maintenance, operation, community training and outreach. It therefore relies on the continuous political will to substantially increase funding for early warning systems for informal settlements in the district. Direct benefits of avoided costs (protection of life in case of an event) should be the strongest argument, but also indirect benefits, which include the increase in hazard knowledge, awareness, response capability, reduced anxiety levels, and community bonding of the inhabitants, should be considered in this decision.

3) socially integrated

The living lab approach reached about 12% of the population in about 40 workshops and community meetings in the course of four years. In addition to this social work, the ground drillings and construction of the monitoring system contributed substantially to community outreach. It was not only the physical construction activity that drew attention to the project, but also the scientific personnel, who had a daily presence for months on end in the field. The academic team formed close bonds with the residents at risk, gained valuable insights into the living conditions and personal challenges of the population and contributed substantially in building trust and acceptance of the pilot project.

Despite the intensive community outreach and prominent installations in public space, a representative survey conducted in 2022 showed that only about half of the residents have even heard of the project. This number seems low at first glance, however, in low-income neighborhoods, other more acute life risks (sudden income loss, bad health, gang activity, etc.) do command more attention than preparing for an eventual landslide that might never happen. Additionally, a high fluctuation of residents and the Corona pandemic during the research phase further aggravated community outreach. Regular community training and information campaigns by experienced social workers are further needed to sustain the LEWS and maintain its acceptance in the community. The long-term success of the LEWS will depend on the sufficient funding of governmental agencies to maintain community training.

A further critical issue for the future acceptance of the LEWS in Bello Oriente is, how well in the perception of residents, the risk management agencies will handle evacuation and eviction of households threatened by an imminent landslide. Current governmental options for the evacuated families  are perceived as not satisfactory up to the point that residents have reacted negatively in the past to DAGRD personnel entering the pilot-neighborhood (the research project has improved that bias).

Therefore, the concern that better landslide monitoring and reaction capacities will result in more evictions has been voiced quite frequently by local committees. If better shelter and housing solutions are not found, residents might perceive the LEWS as an enabler of eviction and reject it in the future.

4) multi-sectoral

The living lab approach developed the LEWS with five stakeholder groups: residents, academics, government, civil society organizations and private companies. A substantial amount of time and effort (about 200 coordination meetings) was spent to engage the various stakeholders in a collaborative process. While collaboration and outreach efforts with the community at risk (40 workshops, 2 emergency drills) were expected to be resource intensive, the collaboration with governmental

stakeholders bound more challenges than expected due to frequent leadership changes, bureaucratic and legal hindrances and not formally allocated budgets to the project. While the voluntary collaboration and perseverance of Medellín's municipal agencies, especially of the municipal risk management agency DAGRD, was exceptional, the continuation of the prototype beyond the research phase was for the longest time not assured. As a bad coincidence, Medellín's premier agency responsible for early warnings (SIATA) was heavily impaired by political infighting shortly after the research commenced. While this

incapacitation was even exceptional for Medellín's political landscape, these unforeseen events are the most important lessons of the living lab methodology. It forced the research team to think about a LEWS that still can offer benefits even under a reduced government support scenario, which can be often the case for municipalities with a high rate of informal settlements (Werthmann, 2021; Perlman, 2010).

While the experimental operation, testing and further development of the LEWS prototype beyond the research phase has been

confirmed by the city of Medellín end of 2022, its continuation has been put on hold at the end of 2023 due to legal constraints. The lessons learnt are (1) that a precise knowledge of legal requirements for academic projects (in this case Colombian donation laws) is necessary before the installation of technical equipment, and (2) that a funding situation where Colombian and German partners are equally financed from the beginning would be strongly preferable. Despite all hindrances, the strong will of the Colombian partners to support the project throughout all phases was decisive and grounded its design

deeper into the political and social realities provided valuable lessons for the future.

5) socio-spatially integrated

The living lab approach integrated the sensor system of the LEWS as miniature public spaces in form of individually painted brick benches and brick cubes, tree plantings, ground markers and information signs. In a representative survey only 17% of

695 the residents were aware that these elements housed the sensors of the LEWS. About 19% actively used and liked them as recreational spaces. These relatively low numbers can be partially explained by the fact that the installations can only be enjoyed in the upper half of the settlement. Overall, the public installations are well used and only in a few cases, the installation of the benches is perceived by some residents as problematic as they can be used by criminal gangs to execute territorial

control. Vandalism of the miniature public spaces is low. These assessments are preliminary and further evaluation is needed to assess the long-term impact of socio-spatial integration.

### 6) multi-scalar

Remote sensing data with spatial resolutions of around 10cm allow to localize the exposed buildings with high precision and indirectly the exposed population. Thus, these data provide crucial quantitative and spatio-temporal information to support the planning of the LEWS. However, these high-resolution data from drones, flight campaigns or now even satellites are very costly in time and resources and thus are not yet an option for large area or even global applications. However, beyond this capability on local levels (i.e. to work on individual building level) lower resolution data free of cost are available globally. They allow - although thematically less detailed and less accurate - to locate exposed areas in terms of topography or built environments or indirectly also population. This enables a city-wide approach to assess where further LEWS can be deployed most effectively, if desired.

### 7) replicable

The living lab developed a prototype of a LEWS which relies on 1) the precise analysis of landslide risk by academic researchers or specialized firms, 2) the risk monitoring through observation by residents and through ground sensors monitored by experts 24/7, 3) the warning dissemination and 4) response by residents and governmental authorities. About 32 sites have been identified in Medellín, where variations of the LEWS could be useful (cf. Sapena et al., 2023). In the moment, a test and calibration phase are conducted in order to investigate the practicability and subsequently the replicability potential of the prototype. As a first insight, it can be stated that this type of LEWS is confined to urban areas where there is sufficient technical expertise to operate the complex sensor-based instrumentation. Secondly, local risk management agencies are worried about the substantially increased effort of operating the proposed system compared to current practices. Increased political will is required to expand the capacities and funding of local risk management authorities for the construction, operation and maintenance of this type of LEWS. Thirdly, risk management authorities will have to find better procedures for housing evacuees. If not, informal residents will reject the LEWS out of fear of eviction.

### 8) redundant

Redundancy has not been listed as a specific goal at the outset of the research as it is a commonly acknowledged strategy for achieving greater resilience in systems especially EWS (Michoud et al., 2013). During the course of the research project, it was necessary to increase aspects of redundancy to maintain basic functions of the LEWS in case of wavering government support or infrastructure failure. Therefore, the low-tech, self-help capacities regarding landslide monitoring, warning dissemination and response capacity of the at-risk community came to the foreground as the first line of defense in case everything else fails; even if budgets are low, community capacity needs to be still fostered as the most relevant ingredient and expense of any LEWS.

## 5 Final Statement

The research has shown that precise, affordable and socially integrated LEWS in informal settlements are not out of reach, but require a substantial increase in effort. Based on the preliminary experiences with the LEWS in Bello Oriente, an adaptation of the system to other locations with different conditions is conceivable as the modular technical components of the LEWS can be down or upgraded as needed. As a first-time project, the technical aspects of the LEWS in Bello Oriente were more time and cost intensive than an off-the-shelf application. While the cost for technical equipment will decrease with more widespread application, the effort for social outreach, collaboration and training of residents will not. As the LEWS is both, a technical and a social enterprise, the two elements need constant care by trained personnel.

Based on these preliminary findings, some fundamental prerequisites for a municipality to start an LEWS in an informal settlement can be formulated:

1) LEWS make only sense in areas where the resettlement of vulnerable residents is infeasible or will last for decades.

2) government authorities and its representatives need to already have or regain the trust of the residents of landslide-prone informal settlements. Trust can be achieved through consistency of policies, steady social engagement over longer time periods or during implementation of an LEWS.

3) the government has to keep a constant open and honest interaction with the civil organizations of the territory since they are key to understand and solve challenges and conflicts with the community.

4) government authorities have to have a long-term commitment independent of election cycles to support an LEWS in informal settlements. Financial allowances and personnel to socially and technically sustain the LEWS have to be allocated long-term in the governmental budget. The sustenance and expansion of low-tech, self-help capacities of the at-risk community has to be an inherent part of this budget.

5) a commonly accepted process of temporary and/or permanent resettlement in case of imminent landslide threat has to be in place (third warning level "Alert"). As it can take days, weeks or even months for the landslide to be triggered, the affected residents need a temporary and/or permanent resettlement solution.

The presented living lab approach is not the first LEWS to be installed in an informal settlement; other attempts have been made, for example in Rio de Janeiro (Heffer and ELLA, 2013), Honduras (Peters et al.; 2022), or in Nepal (Thapa and Adhikari, 2019). The interim results of the living lab confirm the common notion that an LEWS is a predominantly social system with embedded technical elements, even more so in an informal settlement with its strong social and political dynamics (Kelman and Glantz, 2014).

The presented living lab approach expands this socio-technical understanding by a spatial component. The interaction between the technology of the monitoring system (the technical) and the residents and social organizations of the informal neighborhood (the social) has deliberately transformed the physical space of the neighborhood (the spatial). The new miniature public spaces test the notion that the spatial dimension of an LEWS can contribute to the risk awareness of affected residents while improving livelihood. The understanding that the physical presence of an LEWS is more than just an assemblage of black boxes only

begins (Mazereeuw and Yarina, 2017). In our case its usefulness needs to be further evaluated in the coming years. If successful, the deliberate design of risk technology in public spaces could become a further design parameter for LEWS in urban areas.

A second aspect of innovation of the developed LEWS is the technical monitoring of not-yet-active ground movements in an

informal settlement. In Medellín, ground sensors are usually placed only in an active landslide or near critical infrastructure. The preemptive placement of sensors promises to give more precise monitoring capacities, but is also more maintenance intensive and needs a refined process of data interpretation in order not to cause false alarms. Through the connection of the sensors with the app, residents will be able to see the status of sensors in their neighborhood in the future. Therefore, the correct interpretation of the sensor readings and its continuous surveillance is the most important task of the ensuing calibration phase

of the system.

As the living lab approach developed a prototype of an integrative and participatory LEWS that introduces novel technology, additional time for calibration, development and adjustment is needed in order to reach a certain level of functionality. Its overall usefulness is recognizable, but not yet proven in its full operational business, and its continuation by local authorities

is far from assured. It is crucial to scientifically observe and evaluate its further development and trajectory.

**Appendices**

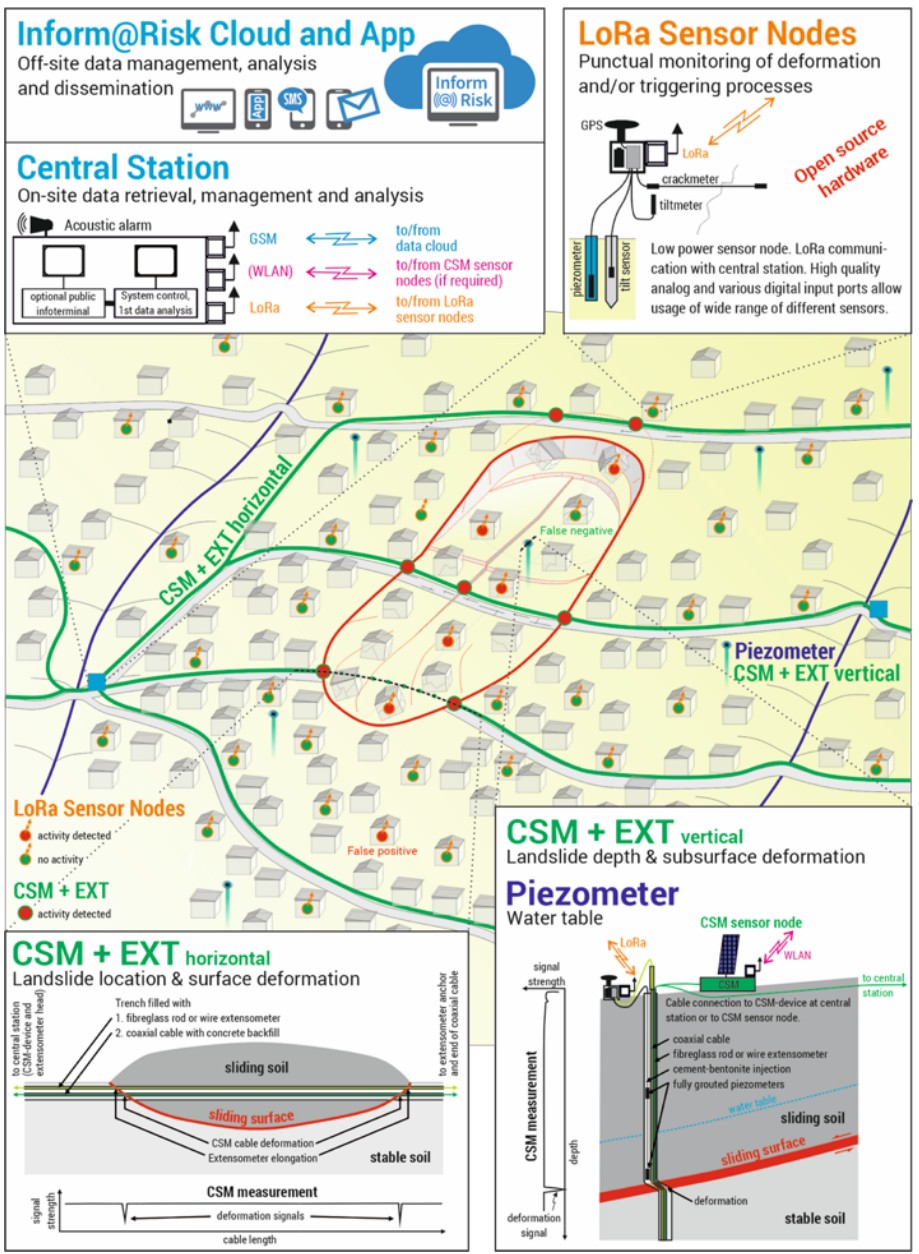

**Figure A1.** Schematic layout of the Inform@Risk sensor-based monitoring system. The system achieves a high spatial density of observations by combining linear deformation measurements (Continuous Shear Monitor and Extensometers) with punctual motion detection based on tilt measurements (wireless (LoRa(R)) measurement nodes). Additionally, the landslide triggering factors rainfall and groundwater table are monitored. The acquired data is processed in a cloud system utilizing methods of sensor fusion to improve the reliability of the system (e.g. only if several sensors indicate a landslide (red dots), an automated public warning is issued). (Thuro et al., 2020, copyright Ernst & Sohn GmbH. Reproduced with permission)

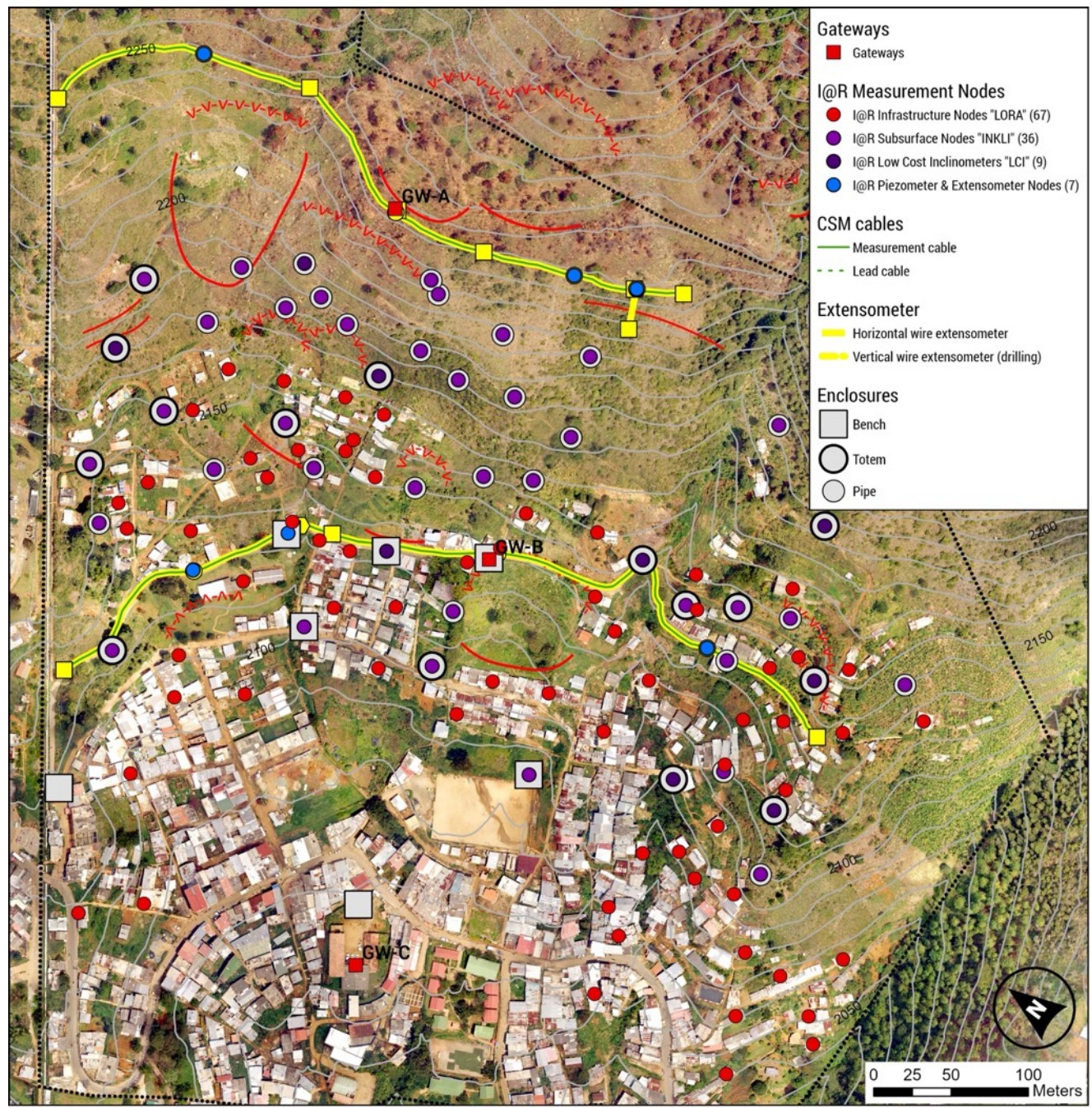

**Figure A2.** Map showing the spatial distribution of the elements of the sensor system in the project area. Red lines indicate landslide phenomena. Background: orthomosaic of drone images acquired in Nov. 2019.

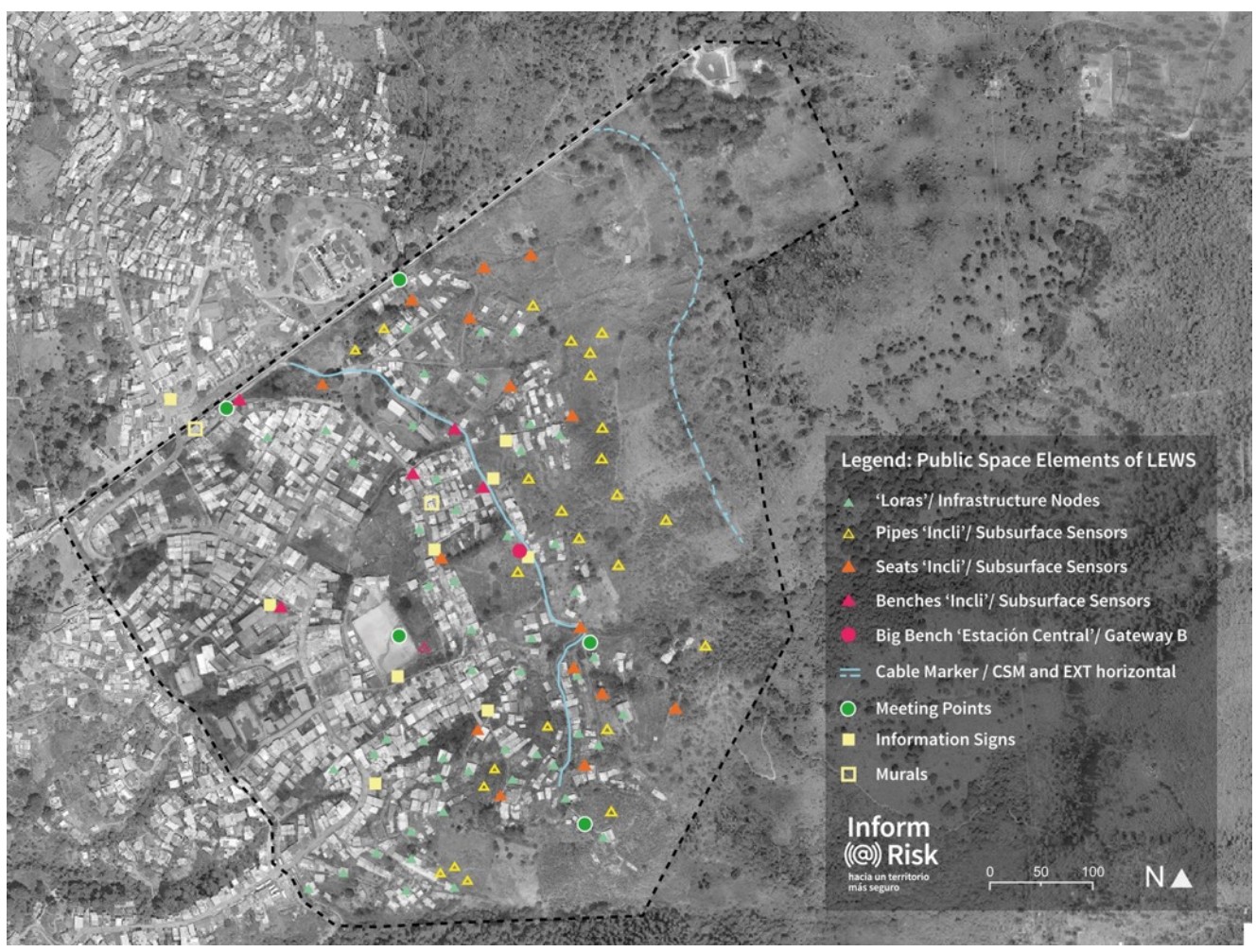

**Figure A3.** Map with location of LEWS elements in public space. Background: orthomosaic of drone images acquired in Nov. 2019.

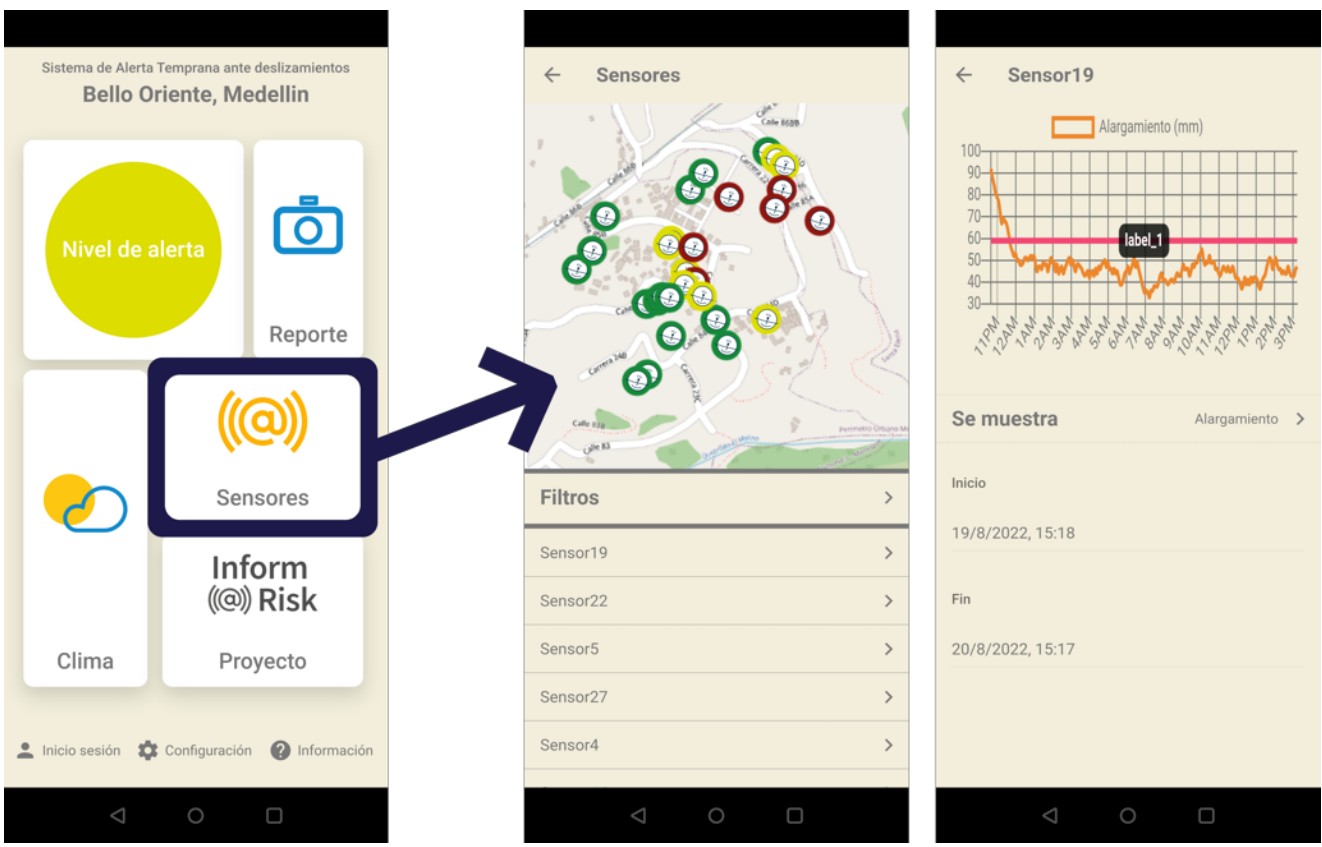

**Figure A4.** Interfaces of the warning app (from left to right): main menu of the App, map of sensor distribution, and information provided by sensor.

## Author Contributions

All authors contributed to the study conception and design. The first outline and final version of the manuscript was developed by Christian Werthmann. In the section "3 Preliminary Findings", the following authors were mainly responsible for the research and content of: the section "3.1.1 Landslide Risk at City Scale" by Marta Sapena and Marlene Kühnl, "3.1.2 Landslide Hazard at Pilot-Neighborhood" by Bettina Menschik, Tamara Breuninger, Moritz Gamperl and Kurosch Thuro. "3.2.1 Automatic Sensor-Based Monitoring" by John Singer, 3.2.2 Manual and People-Based Monitoring by Carolina Garcia, and "3.2.3 Socio-spatial Integration of Monitoring System" by Christian Werthmann and Heike Schäfer. "3.3.1 Warning Levels", "3.3.2 Warning Dissemination channels", "3.3.4 Alert Network" and **3**.4 Response Capacity by Carolina Garcia, the "3.3.3 Warning App" by Sebastian Schröck. The abstract and the section "4 Living Lab Experiences and Conclusions" was a joint effort. All other sections were written by Christian Werthmann. All authors commented on previous versions of the manuscript and all authors read and approved the final manuscript.

## Competing Interests

Dr. John Singer (AlpGeorisk) is in negotiation regarding a contract with the municipal risk management agency of Medellín, DAGRD to continue the development and testing of the sensor system in future. The used data management and analysis platform (AlpGeorisk ONLINE) is developed and commercially distributed by AlpGeorisk.

The other authors have no relevant financial or non-financial interests to disclose.

## Disclaimer

## Special issue statement

## Acknowledgements

This research was funded by the German Federal Ministry of Education and Research as part of the FONA Client II initiative,
grant number 03G0883A-F, Inform@Risk – Strengthening the Resilience of Informal Settlements against Slope Movements. Many thanks go to Ms. Fretzdorff and Ms. Putbrese from Project Management Jülich for their great support and guidance. Thanks go to the Departamento Administrativo de Gestión del Riesgo de Desastres of Medellín (DAGRD) who provided funding for some of the on-site installations and was willing to continue working on the pilot EWS once the Inform@risk research project had ended. All the listed Colombian partners volunteered to invest time and effort. Special thanks go to
Alejandro Echeverrri, director from urbam at EAFIT university and Laura Duarte director from DAGRD for their continuous support of the project. Many thanks go to the residents of Bello Oriente for their support and collaboration.

Living Lab Participants:

German partners: Leibniz Universität Hannover (Coordinator), Technische Hochschule Deggendorf, Technische Universität
München, Deutsches Zentrum für Luft- und Raumfahrt e.V., AlpGeorisk, Sachverständigenbüro für Luftbildauswertung und Umweltfragen;

Colombian partners: EAFIT University - urbam, Departamento , Administrativo de Gestión de Riesgos – DAGRD, Alcaldía de Medellín, Área Metropolitana del Valle de Aburrá, Departamento Administrativo de Planeación, Secretaría de Medio Ambiente, Secretaría de Infraestructura Física, Sistema de Alerta Temprana del Valle de Aburrá, Sociedad Colombiana de
Geología, Colectivo Tejearañas, Corporación Convivamos, Fundación Palomá, Red Barrial Bello Oriente, Residents of Bello Oriente.

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
