# Peer review of "Insights into the Development of a Landslide Early Warning System Prototype in an Informal Settlement: the Case of Bello Oriente in Medellín, Colombia"

_Natural Hazards and Earth System Sciences, 2023_

## Referee Comment (RC2)

**Referee Comment**

1. General comments

The paper is scientifically significant and represents a substantial contribution to the discussion on the socio-cultural dimension of a LEWS. The living lab approach provided valuable insights into the practical challenges involved. The results are discussed in an open, appropriate and balanced way, but sometimes critical findings are not addressed without further consideration of how to deal with them. Overall, the document is well structured but has some inconsistencies (Chapter 3.3). It is suggested that the presentation of the maps be improved to make them easier for the reader to locate.

2. Specific comments
   - (22) "shortage of urban land" is only one of many reasons for informal settlements
   - (31-34) Findings about the technical feasibility and functionality of the LEWS should be mentioned as well.
   - (65) ISO 22327 (2018). Reference to the UNISDR Concept for end-to-end and people-centred warning (4 elements) should be included
   - (85) "affordable" how is this proofed?
   - (89) what is the definition of a socially "integrated LEWS"?
   - (100) what is the difference between "socially integrated" and "socio-spatial integrated"?
   - Figure 1: why is the Pilot Project / EWS only linked to the Government box? The figure does not provide consistent information on the capacities of the actors/stakeholder as indicated in the title.
   - Figure 2: how can the risk assessment be concluded in 10/19 while the vulnerability assessment goes on until 04/21?
   - Figure 2: what is an evacuation system?
   - Figure 3 and 5, 6, 7, 8: it is difficult to get an idea about the location as all maps have different scales and no common references (i.e. outline of the pilot area) which would help for proper orientation
   - Figure 4: Concept of "response"? The figure shows 3 warning levels and dissemination modes but actually no response
   - Chapter 3.3. The chapter title "Warning dissemination" is not appropriate as the chapter goes beyond this concept. It also elaborates on warning thresholds and warning levels, which is part of the warning generation process. On the other hand, a clear description of the warning chain including responsibilities to call for evacuation is not presented.
   - (418) How do you define "short- to medium-term hazard level"?
   - (435-439) as different geographic areas can be affected, how is this reflected in the warning process (in terms of geographically specific warnings)?
   - (440) Affiliation of the expert?
   - Table 2: the red level is somehow inconsistent: the alert level says "prepare for evacuation" while the recommended action is to "evacuate"
   - Chapters 3.3.2, 3.3.3 und 3.3.4 are all somehow about warning dissemination channels. In chapter 3.3.4 the concept and character of the "alert network" is not clear.
   - (484) "Capacity reaction"? Does it mean response capacity?
   - (484) the "possibility of a fast-onset landslide with just minutes of warning time" is not related to the expected increase in monitoring and dissemination capacities of the LEWS prototype.

- 3.4.1. Evacuation planning is usually responsibility of local authorities. How have they been involved?
- (494) "Basic evacuation actions were established for each warning level." According to the warning scheme presented in Table 2, evacuation is only required for the red level.
- (511): "One of the unresolved challenges that remains is the best timing of evacuation…". This issue touches one of the core challenges for landslide EWS. If not solved appropriately, the usefulness of the whole system is in question.
- (540-554) Conclusions on "precise": the main question is whether the "to date unparalleled precision in terms of location and level of hazard" can be translated into reliable and timely warnings and hence into appropriate action. There are obviously still major challenges in this regard (see comment on 511).
- (545-565) Conclusions on "affordable": the chapter provides information on the cost of the monitoring system, but no indication about the costs for "the medium to long-term running costs (maintenance, operation and community training)". It also doesn't considerate the relatively extensive hazard assessments and drillings nor the substantial amount of time and effort (about 200 coordination meetings) spent to engage the various stakeholders in a collaborative process" (592).
- (545-565) Conclusions on "socially integrated": (569): "A low vandalism rate (after six months) indicates a satisfactory level of social acceptance" seems to refer to 5 "spatially integrated" (609).
- (589-607) Conclusions on "multi sectoral": it seems that the ownership of the system is not yet that clear and assured. This is a critical issue and possibly related to the set-up of the cooperation, where research and piloting approaches (with funding and an experimental goal) meet implementation issues (without proper funding and unsteady political commitment)…
- (630-641) Conclusions on "replicable": relatively extensive hazard assessments, drillings, application of detailed hydrogeological and geotechnical models, which have been calibrated by observational data from hydrogeological field tests, geotechnical laboratory tests (291) does not sound like an easy replicable system…
- (642-650) Conclusions on "redundant": "Therefore, the low-tech, self-help capacities regarding landslide monitoring, warning dissemination and response capacity of the at-risk community came to the foreground as the first line of defense in case everything else fails" seems to be a valid conclusion, but need to be integral part of the LEWS approach. How this can be achieved, has not been visualized so far.
- Final statement: It is stated that the usefulness of the prototype is not yet proven and requires long term operation of the system, observation of parameter, adjustments, further developments and continuous interaction with the residents of the informal settlement which should be taken over by the local disaster risk management organisation. In view that this requires appropriate funding, specialized technical expertise and commitment, the authors leave it pretty much open how and whether this can be achieved and how future findings will be taken up by the scientific community.

3. Technical corrections
   - No observations

---

## Author Response (AR1)

**Responses to Reviewers Comments for:**

Insights into the Development of a Landslide Early Warning System Prototype in an Informal Settlement: the Case of Bello Oriente in Medellín, Colombia

**List of Relevant Changes**

based on reviewers comments

Title: changed to better reflect content

Abstract: augmented to better reflect findings

1 Introduction

More detailed explanation of governmental risk management structure of Medellin

More detail and embedment of research goals into existing literature

2 Methodology and Process

Figure 1,2, 3 and 4 revised, based on reviewers comments

3 Preliminary Findings

3.1. Landslide Risk Analysis:

Revision of population numbers

Figure 6, 7 and 8 revised, based on reviewers comments

Additional sources

3.2 Landslide Monitoring

Minor text revisions

3.3 Warning

Changed titles of chapter

More detail on warning levels, warning dissemination channels and warning network

Table 2 minor revision of warning levels

**Individual Responses to Reviewers Comments**

Reviewers comments are in red, our answers in blue

**REVIEWER 1**

A very nice paper, well structured and amazing state of the art being achieved.

Thank you very much for the positive reaction.

My only main critique is that the paper reads like a summarised descriptive project report. Maybe it would be goo to add guiding research questions in the beginning and discuss later on which of these have been achieved by which methodology or scientific approach.

We propose to strengthen the sections "1 Introduction" and "4 Living Lab Experiences and Conclusions" in order to make the underlying research questions and its results clearer. In section 1 we introduce the seven goals that the multidisciplinary research team identified as research objectivesWe will make the research questions that guide the goals clearer in this section by adding additional research context (more references and explanations). In section 4 where we revisit these goals, we will precise our findings in more detail. We believe that this will strengthen the embedment of this particular case and guiding questions/goals into the larger context of early warning system research. As this paper tries to summarize, synthesize and discuss the findings of several disciplines working on specific research questions in a large inter- and transdisciplinary research project, we will not venture too deep into the detailed research questions of the individual teams. The individual disciplines in the research project have published or will publish their disciplinary findings in separate papers with greater detail. Our goal for this paper is to enmesh the

findings and relate them to existing knowledge.

Regarding the methodology: while the overarching research followed the living lab methodology, other methodologies by the disciplinary teams were used to produce specific findings without being specifically described in section "2 Methodology and Process" in order to cut on length of the paper. Detailed descriptions of methodological approaches can be found in the disciplinary papers of individual teams. However, we added a paragraph in section 2 to clarify methodology and augmented section 3 "Preliminary Findings" with additional methodological information.

In summary the additions in section 1, 3 and 4 should embed the research in a larger context and make the research goals more transparent.

Other than that, just some minor comments below.

Inform@Risk in the title is not clear enough that this is a project acronym. It could be confused with a commercial advertisement, so I suggest deleting it in the title.

We agree and will delete "Inform@Risk" in the title.We also propose a new title: "Insights into the Development of a Landslide Early Warning System Prototype in an Informal Settlement: The Case of Bello Oriente in Medellín, Colombia"

Page 2, line 60 spell out LEWS earlier on in the introduction

corrected

Page 3, 75: this part of the sentence needs more explanation: "especially in areas with critical infrastructure." Why are there more sensors in theses areas (because geosensors allow better monitoring ?!?) and what do the authors understand under CI?

Yes, the city of Medellin has geosensors for a more precise monitoring of soil movement close to aqueducts, water tanks, etc. We will modify the sentence to: "In several cities, including Medellín, geosensors that allow a higher precision of monitoring of earth movements in real time are placed in active mass movements (Michoud et al., 2013), especially in areas with critical infrastructure such as aqueducts, water tanks, oil pipes, main roads, etc."

Why did the project have these goals? Following literature, or selected by experts or...?

The goals were selected based on available literature on early warning systems, expert discussion in the research group, investigation into the risk management practices in Medellín and prior urban research in the informal settlements of Medellín. We will augmented the sentence in line 71 to: "Based on literature review and expert discussion, seven goals were set to develop an integrated LEWS that is ..."

We also added additional references in the detailed goal descriptions (see also answer to first comment)..

Fig1: How were these capacities derived? Literature? Project internal discussion... Add sentence on explanation

We plan to adjust the caption to:

"Figure 1. Capacities of actors in the research project. The composition is based on key actors in early warning systems (UN-ISDR, 2006; Fathani et al, 2016) and adjusted to the specific composition of available stakeholders in Medellín."

Page 14, 264: can you kindly add a source or is this maybe "common expert jargon"?

The "Fahrböschung", which was introduced by Heim (1932) is commonly used in Geology to characterize and empirically predict the runout of landslides. The Fahrböschung is the angle between a horizontal plane and a line from the top of a rockfall source scar to the stopping point for any given landslide. It is important that the line follows the track of the landslide. We will add the according reference.

**REVIEWER 2** Harald Spahn

Referee Comment

1.   General comments

The paper is scientifically significant and represents a substantial contribution to the discussion on the socio-cultural dimension of a LEWS. The living lab approach provided valuable insights into the practical challenges involved. The results are discussed in an open, appropriate and balanced way, but sometimes critical findings are not addressed without further consideration of how to deal with them. Overall, the document is well structured but has some inconsistencies (Chapter 3.3). It is suggested that the presentation of the maps be improved to make them easier for the reader to locate.

*Thank you for the overall appreciation and the detailed and very helpful comments. We plan to improve the discussion of findings, correct chapter 3.3 and re-work the maps. Specific improvements can be found in our answers to "2. Specific Comments".*

2.   Specific comments
(22) "shortage of urban land" is only one of many reasons for informal settlements

*We agree: lack of affordable housing, lack of appropriate financing mechanisms, lack of urban planning, lack of political will to develop housing for low-income populations, etc. could be listed as reasons as well.  In order to keep the abstract concise and dwell not too much on the myriad reasons for informal construction, we propose to omit "due to shortage of urban land". The sentence could read: "Self-constructed or informal settlements are frequently built in hazardous terrain such as landslide-prone slopes."*

(31-34) Findings about the technical feasibility and functionality of the LEWS should be mentioned as well.

*We will expand the abstract  in this regard and change to something like:*
*"First findings indicate that the integrative development of technical aspects of a LEWS in informal settlements can be challenging, but manageable; whereas, the social and political support is beyond the control of the designer. Steady political will is needed to increase technical capacities and funding of the operation and maintenance of an increased amount of monitoring equipment. Social outreach has to be continuous in order to inform, train, maintain trust and increase the self-help capacities of the often rapidly changing population of an informal settlement. Satisfying replacement housing options for the case of evacuation have to be in place in order to not loose the overall acceptance of the LEWS.  As political will and municipal budgets*

*can vary, a resilient LEWS for informal settlements has to achieve sufficient social and technical redundancy to maintain basic functionality even in a reduced governmental support scenario."*

(65) ISO 22327 (2018). Reference to the UNISDR Concept for end-to-end and people- centred warning (4 elements) should be included

*The following sentence will be inserted:*
*"The presented findings follow the four-level structure of early warning systems of UNISDR (UNDRR, 2023)."*

(85) "affordable" how is this proofed?

*The idea of the LEWS was to design a prototype which takes advantage of contemporary digital monitoring technologies and still is affordable in an emerging economy such as in Colombia or nations with comparable means. As the prototype was closely developed with the local disaster risk management agency, there was a regular feedback and discussion of expected cost of installation and operation.*
*Nevertheless, a final proof of affordability cannot be given yet, since the prototype is still in a test and calibration mode by the municipality of Medellin and has not been fully integrated into the standard procedures. Final operation costs (and therefore affordability) are still unclear.*

*Affordability is further discussed in the section "conclusions", where the pertaining comments of this review will be further addressed. At this beginning section of the paper, we propose to rephrase the sentence as follows:*

*"2) affordable*
*Precise landslide monitoring systems are available, but they can be cost-intensive. Therefore, the goal is to develop a* **more precise LEWS in accordance with the financial means of an emerging economy***. Affordability shall be increased in materials, installation and maintenance by using cost-effective Internet of Things solutions for the sensor system (e.g. Prakasam et al., 2021; Esposito et al., 2022)."*

(89) what is the definition of a socially "integrated LEWS"
(100) what is the difference between "socially integrated" and "socio-spatial integrated"?

*These two comments can be addressed in tandem:*

*The goal (and necessity) of social integration results from experiences with participatory warning systems (Baudoin et al., 2016; Marchezini et al., 2018), but also stems from ISO 22327 for a community-based landslide early warning system. Particularly in ISO 22327 it is described how the vulnerable population is integrated into seven subsystems of an LEWS (socioeconomic or culture survey, dissemination and communication of knowledge, establishment of a disaster preparedness team, monitoring, early warning and evacuation drill, etc).With the term "socially integrated", we tried to summarize all these participatory activities.*

*The need for socio-spatial integration results from the observation that built objects in informal settlements can be rejected by the population when they have not been installed in a participatory process and with a practical purpose. Accordingly socio-spatial integration goes beyond the mere spatially appropriate placement in the neighborhood. One could argue that the socio-spatial*

*integration of monitoring equipment is a subset of social integration of the LEWS and could be subsumed under that goal. As the research project placed a focus on spatial integration as an experiment for risk communication and public space improvement we decided to list it as a separate point.*

Figure 1: why is the Pilot Project / EWS only linked to the Government box?

*The term pilot project in this diagram refers to Medellín's multi-hazard early warning project SIATA which is a pilot project in itself separate from "our pilot project". This is obviously confusing. We will omit the term in the diagram.*

The figure does not provide consistent information on the capacities of the actors/stakeholder as indicated in the title.

*We will change the figure title to:*
*"Figure 1. Capacities of actors in the research project. The composition is based on key actors in early warning systems (UN-ISDR, 2006; Fathani et al, 2016) and adjusted to the participating stakeholders in Medellín."*

Figure 2: how can the risk assessment be concluded in 10/19 while the vulnerability assessment goes on until 04/21?

*Thanks for pointing this out. The graphic will be updated to show the correct dates.*

Figure 2: what is an evacuation system?

*We agree that the term is confusing, so we removed the words "Evacuation System" from the figure and added terms that references the terminology used in the text.*

Figure 3 and 5, 6, 7, 8: it is difficult to get an idea about the location as all maps have different scales and no common references (i.e. outline of the pilot area) which would help for proper orientation

*All Figures mentioned will be corrected. The project area will be marked in Figures 3 and 5, and scale bars added for better orientation. Figures 6, 7 and 8 will be displayed on the same scale.*

Figure 4: Concept of "response"? The figure shows 3 warning levels and dissemination modes but actually no response

*Figure 4 will be corrected.*

Chapter 3.3. The chapter title "Warning dissemination" is not appropriate as the chapter goes beyond this concept. It also elaborates on warning thresholds and warning levels, which is part of the warning generation process.

*We renamed the chapter as "Warning" in order to include all aspects described in the chapter.*

On the other hand, a clear description of the warning chain including responsibilities to call for evacuation is not presented.

*As explained in the text, the residents were the ones who decided not to name specific persons in charge of the warning dissemination, but to promote the involvement of the whole community. We also added a paragraph to explain why we promote the use of a warning network, instead of a warning chain, where the failure of one single link can jeopardize the whole LEWS. Additionally, we clarified that a detailed description for each warning level, including not only the qualitative and quantitative thresholds, as well as the responsibilities and expected detailed actions for each level was elaborated and distributed to the community.*

(418) How do you define "short- to medium-term hazard level"?

*In context of the warning levels the medium-term hazard level (pre-warning) is about 1 week to 1 month and is based on previous accumulated rainfall (2 weeks / 4 weeks), current groundwater levels and the weather prediction and long-term seasonal climate data.*
*The short-term hazard level is based on current data of the sensor system concerning groundwater and deformation information. This is the basis for issuing warnings concerning the situation in the next hours to days.*
*We will clarify this in the final document.*

(435-439) as different geographic areas can be affected, how is this reflected in the warning process (in terms of geographically specific warnings)?

*The speaker system is used not only to make an alarm sound, but also to send a predefined message that contains the name of the sector of the potential landslide. Also the area affected by a landslide will be presented in the App. An additional sentence will be added under "3.3.2 Warning Dissemination Channels" explaining this fact.*

(440) Affiliation of the expert

*It will be an expert from Medellín's risk management agency DAGRD. This will be added in the manuscript.*

Table 2: the red level is somehow inconsistent: the alert level says "prepare for evacuation" while the recommended action is to "evacuate"

*The red level of the table will be corrected with the following text: "The signs and/or changes in the slope and the weather indicate that a landslide is imminent, evacuate immediately."*

Chapters 3.3.2, 3.3.3 and 3.3.4 are all somehow about warning dissemination channels.
In chapter 3.3.4 the concept and character of the "alert network" is not clear.

*Indeed this can be confusing. We will subsum chapter "3.3.3 Warning App" under chapter "3.3.2 Warning Dissemination Channels" as the app is just another dissemination channel next to sirens and loudspeakers. We will rename chapter "3.3.4 Alert Network" to "3.3.3 Warning Network" and a further explanation will be included:*
*"In order to disseminate the warnings, the project favored the introduction of warning networks. A network concept was preferred by the governing authority and the residents in favor of a warning chain, where the failure of one link can disrupt the whole warning chain."*

(484) "Capacity reaction"? Does it mean response capacity?

*Correct, it should mean "response capacity".*

(484) the "possibility of a fast-onset landslide with just minutes of warning time" is not related to

the expected increase in monitoring and dissemination capacities of the LEWS prototype.

*We will clarify this seeming contradiction by rewriting the sentence to:*
  *"The expected increase in monitoring and dissemination capacities of the LEWS prototype should lead to more timely responses. However, response capacities still have to include the possibility of a fast-onset landslide with just minutes of warning time due to seismic activity or extreme weather events."*

3.4.1. Evacuation planning is usually responsibility of local authorities. How have they been involved?

*In this chapter we explained that the first respondents for evacuation are the community members themselves because of the distant location of the community. The concept of evacuation was developed in accordance with the local authorities. We will includ an additional line to explain this fact:*
*"Basic actions were established for each warning level together with the local community, the risk management authorities and the civil society organizations."*

(494) "Basic evacuation actions were established for each warning level." According to the warning scheme presented in Table 2, evacuation is only required for the red level.

*Correct. The word "evacuation" will be removed in that sentence.*

(511): "One of the unresolved challenges that remains is the best timing of evacuation…". This issue touches one of the core challenges for landslide EWS. If not solved appropriately, the usefulness of the whole system is in question.

*There are multiple factors that play a significant role in the best timing of evacuation, unfortunately many go beyond the capacities of the research project and fall into the responsibility of the local risk management agency. Therefore, we will include a more detailed explanation in the manuscript:*

 *"Due to geo-physical and governmental constraints, one of the still unsolved challenges is the best timing of evacuation. Landslides in the pilot-neighborhood can be either sudden or slow on-set. Therefore, a mass movement might take hours, days, weeks or months, to completely move after first signs have been detected. In addition, anthropic landslide activation plays a major role in Medellín (Alcántara-Ayala and Oliver-Smith, 2019). A landslide with a naturally slow behavior might be suddenly accelerated by human activity (leaking of pipes or septic tanks, slope cutting, insufficient maintenance of drainage channels, etc.) rendering warning times for evacuation extremely short. Accordingly, the research project developed multiple workshops for residents to increase the awareness of correct water management and construction practices, as well as the importance to be prepared to evacuate with very short notice. It is hoped that in the future the training of geo-sensors during and beyond the test phase will precise the quantitative thresholds of triggering factors of mass movements. Thus, the issuance of an evacuation order might become more accurate in relation to landslide activity.*

*Beyond timing, another serious challenge of evacuation lies in the lack of proper housing replacement options for evacuees. Once an evacuation order is issued, it is unclear for evacuees, when and if they ever can return to their homes. Residents in informal settlements rarely have private options (neighbors or friends) to evacuate to for longer time periods. Public agencies are not allowed to build temporary shelters before a landslide has actually occurred. Rent subsidies for evacuees are complex to obtain, can take months to process, are short-lived (three to six months) and are in general too scant for low-income families to afford a new location (Mesa Interbarrial de desconectados, 2011; Rojas et al. 2022). This means that many families are currently fearing an*

*evacuation order and are reluctant to leave their homes, once an evacuation order has been issued. Better housing replacement options need to be developed by the government not only in order to increase compliance with evacuation orders, but also to assure resident's acceptance of an early warning system that they eventually might misperceive as a covert governmental tool for eviction."*

(540-554) Conclusions on "precise": the main question is whether the "to date unparalleled precision in terms of location and level of hazard" can be translated into reliable and timely warnings and hence into appropriate action. There are obviously still major challenges in this regard (see comment on 511).

*We agree, the prototype of a LEWS is still in development. To make that clearer, we will augment the sentence: "Currently, the warning generation procedure and response is in development by the responsible risk management agency DAGRD and still has to prove its capability in the coming months and years." Please see also answer to 511 evacuation.*

(545-565) Conclusions on "affordable": the chapter provides information on the cost of the monitoring system, but no indication about the costs for "the medium to long-term running costs (maintenance, operation and community training)". It also doesn't considerate the relatively extensive hazard assessments and drillings nor the substantial amount of time and effort (about 200 coordination meetings) spent to engage the various stakeholders in a collaborative process" (592).

*The research team intensively re-visited and discussed the issue of affordability, especially in terms which conclusions at this point can be drawn, where the system is still in test mode. The team decided to precise information on affordability in the description of goals in the introduction.*

*In chapter "4 Living Lab Experiences and Conclusions" we decided to rewrite the whole paragraph, it will read:*

*"The prototype of the LEWS aimed to develop state of the art technology with the lowest cost possible in order to be generally affordable and thus be easily replicable. Unfortunately, a conclusive understanding of cost and affordability cannot be given yet as the prototype is still in a test mode by the municipality of Medellín and too many unknowns remain.*

*The costs expanded for the LEWS prototype by the German research team are not representative for the costs of future implementations as they include development costs, German administrative costs and German salaries. It also is not possible to assess the costs for the hazard and risk analyzes and social work in a generalized way as they greatly depend on the complexity of the situation in the individual project site and the level of preexisting know-how and administrative structures, which even within a single municipality can vary greatly. The development of the LEWS prototype required a large amount of coordination with various stakeholders (200 meetings) and intense community engagement (40 workshops). A mature and replicable LEWS might require less coordination meetings in the future and community outreach might decrease. But this will depend on the structure of the community at-risk and the capacities of the risk management authorities in charge. Also, the cost for monitoring the sensor system day and night will vary considerably, if the system is added onto an already existing citywide multi-hazard sensor-monitoring system (like SIATA in Medellín) or if a stand-alone observation unit just for landslides has to be funded.*

*The only component of the LEWS, where sound cost estimates are possible based on our current experience is the installation of the sensor-based monitoring system. Sapena et al. (2023) have performed a study on the cost for implementing the wireless geosensor network in different high-risk areas in Medellín which results in costs between € 5 and € 41 per inhabitant. The cost variation thereby is mainly related to differences in the population density and the required monitoring*

*precision which was varied based on the level of landslide risk in the area.*

*In the end, only future long-term experience will allow to give sound numbers concerning the overall costs of implementing and maintaining the prototype LEWS. However, it is already obvious that the developed prototype with its increased efforts for risk analysis, monitoring and social work requires a considerable budget and trained personnel for installation, maintenance, operation, community training and outreach. It therefore relies on the continuous political will to substantially increase funding for early warning systems for informal settlements in the district. Direct benefits of avoided costs (protection of life in case of an event) should be the strongest argument, but also indirect benefits, which include the increase in hazard knowledge, awareness, response capability, reduced anxiety levels, and community bonding of the inhabitants, should be considered in this decision.”*

(545-565) Conclusions on “socially integrated”: (569): “A low vandalism rate (after six months) indicates a satisfactory level of social acceptance” seems to refer to 5 “spatially integrated” (609).

*We will omit this sentence as it is confusing.*

(589-607) Conclusions on “multi sectoral”: it seems that the ownership of the system is not yet that clear and assured. This is a critical issue and possibly related to the set-up of the cooperation, where research and piloting approaches (with funding and an experimental goal) meet implementation issues (without proper funding and unsteady political commitment)…

*The ownership has been cleared. The prototype has been transferred to the city of Medellin in December 2022 and the city tests and further develops the prototype as a pilot project. If the pilot project will be fully operational part of the EWS system in the municipality remains to be seen. We will augment (in red) the sentence: “In the end, the experimental operation, testing and further development of the LEWS prototype beyond the research phase has been confirmed by the city of Medellín end of 2022.”*

(630-641) Conclusions on “replicable”: relatively extensive hazard assessments, drillings, application of detailed hydrogeological and geotechnical models, which have been calibrated by observational data from hydrogeological field tests, geotechnical laboratory tests (291) does not sound like an easy replicable system…

*Much of the required data for the analyses is achieved during the hazard and risk assessment, which in any case is a requirement for an effective area wide landslide early warning system. Additionally, the proposed sensor system can also be used without the complex data analysis methods proposed - especially if it is used as a site specific monitoring system after a landslide has been recognized / detected.*

(642-650) Conclusions on “redundant”: “Therefore, the low-tech, self-help capacities regarding landslide monitoring, warning dissemination and response capacity of the at- risk community came to the foreground as the first line of defense in case everything else fails” seems to be a valid conclusion, but need to be integral part of the LEWS approach. How this can be achieved, has not been visualized so far.

*In the research process, there were many workshops with residents that focused on capacity building regarding monitoring, warning and response. This community involvement has been described in the manuscript in chapter “3.2.2 Manual and People-Based Monitoring”, “3.3.2*

*Warning Dissemination Channels", "3.3.3 Warning Network" , "3.4.1 Evacuation Planning"*
*and "3.4.2 Evacuation Training". Based on your comment and to make this involvement clearer,*
*we will add some more details in some of these chapters.*

Final statement: It is stated that the usefulness of the prototype is not yet proven and
requires long term operation of the system, observation of parameter, adjustments, further
developments and continuous interaction with the residents of the informal settlement which
should be taken over by the local disaster risk management organisation. In view that this requires
appropriate funding, specialized technical expertise and commitment, the authors leave it pretty
much open how and whether this can be achieved and how future findings will be taken up by the
scientific community.

*This comment is correct, the presented paper only describes the gained knowledge of a prototype*
*that is still in development. In December 2022, the prototype has been transferred to Medellin's*
*risk management agency DAGRD. Since then DAGRD maintains, tests and further develops the*
*prototype with its own personnel and technicians. However, there is no guarantee that Medellín's*
*government will continue the prototype for the coming years or will alter it substantially fitting to*
*capacities and budgets (for example abandon the sensor system).*
*We strongly agree that it would be important that the further development of the prototype by the*
*municipality should be scientifically observed and documented. Currently no follow up funding*
*could be secured so far. Despite that a core team of the research project stays in regular*
*communication with DAGRD in a consultative role and hopes to publish further results.*
*We propose to make the developmental aspect more transparent by renaming the title of the paper*
*to: "Insights into the Development of a Landslide Early Warning System Prototype in an Informal*
*Settlement: The Case of Bello Oriente in Medellín, Colombia"*

*Furthermore, we would like to alter the last sentence in the final statement:*

*"The continued operation and further development of the system by the risk management*
*agencies of Medellín is still ongoing and its results are still unknown. It is crucial  to*
*scientifically observe and evaluate its further development and trajectory."*

3. Technical corrections
    No observations

**REVIEWER 3**
**GENERAL COMMENTS**

In their manuscript, the authors summarize the preliminary findings of a living lab project for the
implementation of a landslide early warning system in the community of Bello Oriente in Medellín,
Colombia. The paper presents an impressive account of different dimensions of challenges that come
with implementing a landslide early warning system (LEWS), addressing a timely and relevant topic.

It is well written in good English with a straight-forward structure. I particularly appreciate the open
and critical way in which the authors discuss the challenges, successes, and failures of the project.
Please find below some more particular comments, questions, and suggestions for the modification
of the manuscript.

I would recommend the publication of the manuscript after these comments have been addressed,

revisions have been made, and especially the maps have been improved as suggested below.

**SPECIFIC COMMENTS**

**Lines 50 and 52:** EWS and LEWS should be introduced here.

Additional information will be included.

**Line 56 and others:** How is the project related to the local disaster risk reduction (DRR) organizations, such as SIATA, DAGRD, etc.? Are there any redundancies between the project and existing systems and/or what is the gap where your project fits into these local structures?

We will include the following text in the manuscript to clarify the above:

"In the Aburrá valley, where Medellin is located, the regional environmental agency Area Metropolitana Valle de Aburrá (AMVA) operates the research project SIATA. SIATA monitors environmental conditions like weather, hydrology, air quality, seismic activity, among others. SIATA develops forecasts and can send out early alerts to AMVA and Medellin's risk management agency DAGRD, which is responsible for response on the ground. DAGRD, AMVA and SIATA collaborate to monitor active landslides that compromise relevant infrastructure in specific locations, but there is a general need for monitoring large high hazard areas with socially integrated early warning systems. This applies specially to the informal settlements that are located in the high hazard slopes of the rural-urban border of Medellín where most population is highly vulnerable and therefore high risk prevails."

**Lines 72 and 100:** "spatially" or "socio-spatially" integrated?

we will correct line 100 to "socio-spatially" integrated?

**Line 175:** What is the name of the risk management authority of the city of Medellín?

Medellín's risk management agency DAGRD has now been introduced in chapter 1 Introduction (see also answer to Line 56).

**Section 3.1.1, e.g. line 194:** How exactly did you "map" the landslide susceptible areas based on the POT 2014? Are there no other previous works by other authors that analyzed landslide hazard or susceptibility in Medellín that you considered or that are worth mentioning?

The landslide hazard map within the POT 2014 (land use plan 2014) includes all available information on landslide hazard for Medellín until 2014. It is an official map into which hazard maps from the POT 2006 and the National University of Colombia 2009, mass movement inventories from the Administrative Department of Disaster Management, morphodynamic process maps, and all geotechnical and slope stability studies carried out for the municipality since 2006 as well as local knowledge by experts have been incorporated. It is the only map for Medellín, which includes not only theoretical studies, but also local knowledge, which enriches the map especially in the rural parts of the municipality. An updated version of the map within the next land use plan is currently being worked on, but not yet available. To address your very valid concern, we will rewrite the

sentence according to the following:

"On the one hand, the landslide hazard map from the official land use plan (Plan de ordenamiento territorial, POT 2014), which includes the zoning of landslide hazards and combines all available risk-related maps, mass movement inventories, geotechnical and slope stability studies and local knowledge (Alcaldía de Medellín, 2014), was used to map landslide susceptibility areas across entire Medellín."

**Section 3.1.2:** Who did the geological field mapping? Foreign geologists or also local geologists? I understand that for the project a very detailed assessment was required, but were there no geotechnical maps available as a reference?

The geological and geomorphological field mapping was performed by German engineering geologists on the PhD candidate and the bachelor degree levels (from TUM). Unfortunately no small scale geological or geotechnical maps were available.

**Lines 260 to 264:** Could you provide some references for these approaches?

The approach to combine field observations (existing landslides and soil cover thickness) with the critical slope angle (determined using SLIDE-2D and shear tests) is a method used by FOEN (Swiss Federal Office of the Environment). A reference will be given: (FOEN, 2016).

As well as for the term "fahrböschung": (Heim, 1932; Evans & Hungr 1993).

**Line 307:** What is MEMS?

In MEMS (Micro-Electro-Mechanical-Systems) electonics and micro-mechanical elements are combined in a single system/chip. This allows to produce fully functional sensors (e.g. for acceleration or magnetic field) within a single chip with a very small size and often low power consumption. MEMS sensors are used widespread in electronic devices (smart phones, wearables) and thus are mass produced at comparable low cost.

In the text we will add "(Micro-Electro-Mechanical-System)" after MEMS to make the acronym clear.

**Line 381:** What exactly are micro gardens? Are they for growing food?

Some of the residents who agreed to be godfathers of the sensors installed small fenced flower beds next to the sensors. We will augment the text: "micro gardens in the form of small flower beds ..."

**Lines 451 and 452:** Who operates the alarm system and takes the decisions?

We will add the following sentence in the manuscript:

"At the time of writing, the speaker system is operated by community members, who were trained in the use of the instruments. In the near future it is expected to install a control to be able to activate the system remotely by the risk management authorities."

**Lines 541 to 554:** Will the warnings be based on the sensor signals only? Or do you plan to have local experts or residents check the situation on the ground before issuing a warning?

When a sensor displays unusual activity (or a residents reports a problem) an expert from the risk management agency DAGRD will check in the field, before a warning is issued. Only when there is a very fast onset landslide detected by several sensors, it is planned to issue an automatic warning.

Diagram 4 illustrates this procedure. This procedure is also described between line 430 and 435.

**Lines 590 and following, lines 657 and following, and others:** You emphasize the effort for coordination with the different stakeholders and especially the work with the community residents. Did you have any local project members working on the ground permanently? If so, how important were they for the success of the project?

Thanks for pointing this out. There was actual a strong contribution by scientific personnel on the ground. We will add the following paragraph to address the questions:

"In addition to this social work, the ground drillings and construction of the monitoring system contributed substantially to community outreach. It was not only the physical construction activity that drew attention to the project, but also the scientific personnel, who had a daily presence for months on end in the field. They formed close bonds with the residents at risk, gained valuable insights into the living conditions and personal challenges of the population and contributed substantially in building trust and acceptance of the pilot project."

**Line 670:** Should be point 4), and 5) in line 674?

There was a numbering mistake which will be corrected.

**Figure 3 and 5, 6, 7, 8:** I agree with Reviewer #2 that the different map extents and scales are confusing. An overview map showing the different map outlines or a common reference, such as the outlines of the project area, would be very helpful. Moreover, Fig. 3 has no scale and in Fig. 6 the coordinates are illegible. Why are there "islands" floating around in Fig. 5?

The figure will be corrected accordingly.

**References:**

Basher et al. 2016 and Kühnl et al. 2022 were not quoted in the text

We will put Basher et al. 2016 and Kühnl et al. 2022 in the text in the right place.

Hossain et al. 2019 appears twice

This will be corrected.

Padilla Galicia et al. 2009 is incomplete

This will be corrected.

The references should be sorted alphabetically

This will be corrected.

---

## Author Response (AR2)

**n-hess-2023-53-author_response-version2**
December 4, 2023

A preliminary and still further to be developed prototype of an early warning system against landslides was transferred from the academic team of Inform@Risk to the risk management agency of Medellín (DAGRD) at the end of 2022. DAGRD operated and reviewed the system in the year of 2023. In November 2023 DAGRD informed the academic team that they cannot further operate and develop the system as they cannot legally accept the monitoring instruments of the system as a donation. The consequence is that the final functions of the monitoring system (the development of warning thresholds and subsequently the alert dissemination) cannot be further advanced and the whole warning system remains not functional.

Outlook:
The research team is currently looking for another academic operator in Colombia. There is also the hope that a less strict interpretation of donation laws is held by the next municipal administration which starts on January 1st 2024.

This important development needs to be reflected in the paper as it illustrates the special challenges a living lab can encounter when the results of the research are transferred from the academic into the municipal realm. These experiences can be very helpful for other living labs.

Besides that, additional images references have been added as requested by nhess.

Changes:

Line 30 to 45: abstract is rewritten to reflect the described development

Line 105: better wording of affordability issue

Line 196: source of aerial background for figure 3c was added

Line 213 to 224:  end of chapter "Methodology and Process" has been augmented to reflect recent developments

Line 225: source of cartography for figure 5 was added

Line 520 to 521: end of section "3.3.2 Warning Dissemination Channels" has been augmented to reflect recent developments

Line 610, 633, 701 to 708, 742 to 745: section "4 Living Lab Experiences and Conclusions" has been augmented to reflect recent developments

Line 812, 813: the end of section "5 Final Statement" has been changed to reflect recent developments

Line 831: source of aerial background for figure A3 was added

Line 878: one source was added under "References"

---

## Author Response (AR3)

05 Dec 2023
**Editor decision: Publish subject to minor revisions (review by editor)**
by Jörn Lauterjung

Additional private note (visible to authors and reviewers only):
Please replace line 705 ff: "Despite all hindrances, the strong will of the Colombian partners to support the project throughout all phases was decisive and grounded its design deeper into the political and social realities provided valuable lessons for the future." into "Despite all hindrances, the strong will of the Colombian partners to support the project throughout all phases was decisive and grounded its design deeper into the political and social realities. This provided valuable lessons for the future."

Please replace line 783 ff: "Its overall usefulness is recognizable, but not yet proven in its full operational business, and its continuation by local authorities is far from assured." into "Its overall usefulness is recognizable, but not yet proven in full operation, and its continuation by local authorities is far from assured."

Response:

The requested changes have been integrated into the manuscript.